# Optimal Treatment Regimes for Proximal Causal Learning

**Tao Shen**
National University of Singapore

**Yifan Cui**[*]
Zhejiang University

## Abstract

A common concern when a policymaker draws causal inferences from and makes decisions based on observational data is that the measured covariates are insufficiently rich to account for all sources of confounding, i.e., the standard no confoundedness assumption fails to hold. The recently proposed proximal causal inference framework shows that proxy variables that abound in real-life scenarios can be leveraged to identify causal effects and therefore facilitate decision-making. Building upon this line of work, we propose a novel optimal individualized treatment regime based on so-called outcome and treatment confounding bridges. We then show that the value function of this new optimal treatment regime is superior to that of existing ones in the literature. Theoretical guarantees, including identification, superiority, excess value bound, and consistency of the estimated regime, are established. Furthermore, we demonstrate the proposed optimal regime via numerical experiments and a real data application.

## 1 Introduction

Data-driven individualized decision-making has received tremendous attention nowadays due to its applications in healthcare, economics, marketing, etc. A large branch of work has focused on maximizing the expected utility of implementing the estimated optimal policy over a target population based on randomized controlled trials or observational studies, e.g., Athey and Wager (2021); Chakraborty and Moodie (2013); Jiang et al. (2019); Kitagawa and Tetenov (2018); Kosorok and Laber (2019); Murphy (2003); Qian and Murphy (2011); Robins (1986, 1994, 1997); Tsiatis et al. (2019); Wu et al. (2019); Zhao et al. (2012, 2019).

A critical assumption commonly made in these studies, known as unconfoundedness or exchangeability, precludes the existence of unmeasured confounding. Relying on an assumed ability of the decision-maker to accurately measure covariates relevant to a variety of confounding mechanisms present in a given observational study, causal effects, value functions, and other relevant quantities can be nonparametrically identified. However, such an assumption might not always be realistic in observational studies or randomized trials subject to non-compliance (Robins, 1994, 1997). Therefore, it is of great interest in recovering confounding mechanisms from measured covariates to infer causal effects and facilitate decision-making. A prevailing strand of work has been devoted to using instrumental variable (Angrist et al., 1996; Imbens and Angrist, 1994) as a proxy variable in dynamic treatment regimes and reinforcement learning settings (Cui, 2021; Cui and Tchetgen Tchetgen, 2021b,a; Han, 2023; Liao et al., 2021; Pu and Zhang, 2021; Qiu et al., 2021; Stensrud and Sarvet, 2022).

Recently, Tchetgen Tchetgen et al. proposed the so-called proximal causal inference framework, a formal potential outcome framework for proximal causal learning, which while explicitly acknowledging covariate measurements as imperfect proxies of confounding mechanisms, establishes causal

---

[*]Correspondence to Yifan Cui <cuiyf@zju.edu.cn>

37th Conference on Neural Information Processing Systems (NeurIPS 2023).

identification in settings where exchangeability on the basis of measured covariates fails. Rather than as current practice dictates, assuming that adjusting for all measured covariates, unconfoundedness can be attained, proximal causal inference essentially requires that the investigator can correctly classify a subset of measured covariates $L \in \mathcal{L}$ into three types: i) variables $X \in \mathcal{X}$ that may be common causes of the treatment and outcome variables; ii) treatment-inducing confounding proxies $Z \in \mathcal{Z}$; and iii) outcome-inducing confounding proxies $W \in \mathcal{W}$.

There is a fast-growing literature on proximal causal inference since it has been proposed (Cui et al., 2023; Dukes et al., 2023; Ghassami et al., 2023; Kompa et al., 2022; Li et al., 2023; Mastouri et al., 2021; Miao et al., 2018b; Shi et al., 2020b, 2021; Shpitser et al., 2023; Singh, 2020; Tchetgen Tchetgen et al., 2020; Ying et al., 2023, 2022 and many others). In particular, Miao et al. (2018a); Tchetgen Tchetgen et al. (2020) propose identification of causal effects through an outcome confounding bridge and Cui et al. (2023) propose identification through a treatment confounding bridge. A doubly robust estimation strategy (Chernozhukov et al., 2018; Robins et al., 1994; Rotnitzky et al., 1998; Scharfstein et al., 1999) is further proposed in Cui et al. (2023). In addition, Ghassami et al. (2022) and Kallus et al. (2021) propose a nonparametric estimation of causal effects through a min-max approach. Moreover, by adopting the proximal causal inference framework, Qi et al. (2023) consider optimal individualized treatment regimes (ITRs) estimation, Sverdrup and Cui (2023) consider learning heterogeneous treatment effects, and Bennett and Kallus (2023) consider off-policy evaluation in partially observed Markov decision processes.

In this paper, we aim to estimate optimal ITRs under the framework of proximal causal inference. We start with reviewing two in-class ITRs that map from $\mathcal{X} \times \mathcal{W}$ to $\mathcal{A}$ and $\mathcal{X} \times \mathcal{Z}$ to $\mathcal{A}$, respectively, where $\mathcal{A}$ denotes the binary treatment space. The identification of value function and the learning strategy for these two optimal in-class ITRs are proposed in Qi et al. (2023). In addition, Qi et al. (2023) also consider a maximum proximal learning optimal ITR that maps from $\mathcal{X} \times \mathcal{W} \times \mathcal{Z}$ to $\mathcal{A}$ with the ITRs being restricted to either $\mathcal{X} \times \mathcal{W} \to \mathcal{A}$ or $\mathcal{X} \times \mathcal{Z} \to \mathcal{A}$. In contrast to their maximum proximal learning ITR, in this paper, we propose a brand new policy class whose ITRs map from measured covariates $\mathcal{X} \times \mathcal{W} \times \mathcal{Z}$ to $\mathcal{A}$, which incorporates the predilection between these two in-class ITRs. Identification and superiority of the proposed optimal ITRs compared to existing ones are further established.

The main contributions of our work are four-fold. Firstly, by leveraging treatment and outcome confounding bridges under the recently proposed proximal causal inference framework, identification results regarding the proposed class $\mathcal{D}_{\mathcal{ZW}}^{\Pi}$ of ITRs that map $\mathcal{X} \times \mathcal{W} \times \mathcal{Z}$ to $\mathcal{A}$ are established. The proposed ITR class can be viewed as a generalization of existing ITR classes proposed in the literature. Secondly, an optimal subclass of $\mathcal{D}_{\mathcal{ZW}}^{\Pi}$ is further introduced. Learning optimal treatment regimes within this subclass leads to a superior value function. Thirdly, we propose a learning approach to estimating the proposed optimal ITR. Our learning pipeline begins with the estimation of confounding bridges adopting the deep neural network method proposed by Kompa et al. (2022). Then we use optimal treatment regimes proposed in Qi et al. (2023) as preliminary regimes to estimate our optimal ITR. Lastly, we establish an excess value bound for the value difference between the estimated treatment regime and existing ones in the literature, and the consistency of the estimated regime is also demonstrated.

## 2  Methodology

### 2.1  Optimal individualized treatment regimes

We briefly introduce some conventional notation for learning optimal ITRs. Suppose $A$ is a binary variable representing a treatment option that takes values in the treatment space $\mathcal{A} = \{-1, 1\}$. Let $L \in \mathcal{L}$ be a vector of observed covariates, and $Y$ be the outcome of interest. Let $Y(1)$ and $Y(-1)$ be the potential outcomes under an intervention that sets the treatment to values $1$ and $-1$, respectively. Without loss of generality, we assume that larger values of $Y$ are preferred.

Suppose the following standard causal assumptions hold: (1) Consistency: $Y = Y(A)$. That is, the observed outcome matches the potential outcome under the realized treatment. (2) Positivity: $\mathbb{P}(A = a|L) > 0$ for $a \in \mathcal{A}$ almost surely, i.e., both treatments are possible to be assigned.

We consider an ITR class $\mathcal{D}$ containing ITRs that are measurable functions mapping from the covariate space $\mathcal{L}$ onto the treatment space $\mathcal{A}$. For any $d \in \mathcal{D}$, the potential outcome under a hypothetical

intervention that assigns treatment according to $d$ is defined as

$$Y(d(L)) \stackrel{\triangle}{=} Y(1)\mathbb{I}\{d(L) = 1\} + Y(-1)\mathbb{I}\{d(L) = -1\},$$

where $\mathbb{I}\{\cdot\}$ denotes the indicator function. The value function of ITR $d$ is defined as the expectation of the potential outcome, i.e.,

$$V(d) \stackrel{\triangle}{=} \mathbb{E}[Y(d(L))].$$

It can be easily seen that an optimal ITR can be expressed as

$$d^*(L) = \text{sign}\{\mathbb{E}(Y(1) - Y(-1)|L)\}$$

or

$$d^* = \arg\max_{d \in \mathcal{D}} \mathbb{E}[Y(d(L))].$$

There are many ways to identify optimal ITRs under different sets of assumptions. The most commonly seen assumption is the unconfoundedness: $Y(a) \perp A|L$ for $a = \pm 1$, i.e., upon conditioning on $L$, there is no unmeasured confounder affecting both $A$ and $Y$. Under this unconfoundedness assumption, the value function of a given regime $d$ can be identified by (Qian and Murphy, 2011)

$$V(d) = \mathbb{E}\left[\frac{Y\mathbb{I}\{A = d(L)\}}{f(A|L)}\right],$$

where $f(A|L)$ denotes the propensity score (Rosenbaum and Rubin, 1983), and the optimal ITR is identified by

$$d^* = \arg\max_{d \in \mathcal{D}} V(d) = \arg\max_{d \in \mathcal{D}} \mathbb{E}\left[\frac{Y\mathbb{I}\{A = d(L)\}}{f(A|L)}\right].$$

We refer to Qian and Murphy (2011); Zhang et al. (2012); Zhao et al. (2012) for more details of learning optimal ITRs in this unconfounded setting.

Because confounding by unmeasured factors cannot generally be ruled out with certainty in observational studies or randomized experiments subject to non-compliance, skepticism about the unconfoundedness assumption in observational studies is often warranted. To estimate optimal ITRs subject to potential unmeasured confounding, Cui and Tchetgen Tchetgen (2021b) propose instrumental variable approaches to learning optimal ITRs. Under certain instrumental variable assumptions, the optimal ITR can be identified by

$$\arg\max_{d \in \mathcal{D}} \mathbb{E}\left[\frac{MAY\mathbb{I}\{A = d(L)\}}{\{\mathbb{P}(A = 1|M = 1, L) - \mathbb{P}(A = 1|M = -1, L)\}f(M|L)}\right],$$

where $M$ denotes a valid binary instrumental variable. Other works including Cui (2021); Cui and Tchetgen Tchetgen (2021a); Han (2023); Pu and Zhang (2021) consider a sign or partial identification of causal effects to estimate suboptimal ITRs using instrumental variables.

## 2.2 Existing optimal ITRs for proximal causal inference

Another line of research in causal inference considers negative control variables as proxies to mitigate confounding bias (Kuroki and Pearl, 2014; Miao et al., 2018a; Shi et al., 2020a; Tchetgen Tchetgen, 2014). Recently, a formal potential outcome framework, namely proximal causal inference, has been developed by Tchetgen Tchetgen et al. (2020), which has attracted tremendous attention since proposed.

Following the proximal causal inference framework proposed in Tchetgen Tchetgen et al. (2020), suppose that the measured covariate $L$ can be decomposed into three buckets $L = (X, W, Z)$, where $X \in \mathcal{X}$ affects both $A$ and $Y$, $W \in \mathcal{W}$ denotes an outcome-inducing confounding proxy that is a potential cause of the outcome which is related with the treatment only through $(U, X)$, and $Z \in \mathcal{Z}$ is a treatment-inducing confounding proxy that is a potential cause of the treatment which is related with the outcome $Y$ through $(U, X, A)$. We now summarize several basic assumptions of the proximal causal inference framework.

**Assumption 1.** *We make the following assumptions:*
*(1) Consistency: $Y = Y(A, Z)$, $W = W(A, Z)$.*
*(2) Positivity: $\mathbb{P}(A = a \mid U, X) > 0$, $\forall a \in \mathcal{A}$.*
*(3) Latent unconfoundedness:*
*$(Z, A) \perp (Y(a), W) \mid U, X$, $\forall a \in \mathcal{A}$.*

The consistency and positivity assumptions are conventional in the causal inference literature. The latent unconfoundedness essentially states that $Z$ cannot directly affect the outcome $Y$, and $W$ is not directly affected by either $A$ or $Z$. Figure 1 depicts a classical setting that satisfies Assumption 1. We refer to Shi et al. (2020b); Tchetgen Tchetgen et al. (2020) for other realistic settings for proximal causal inference.

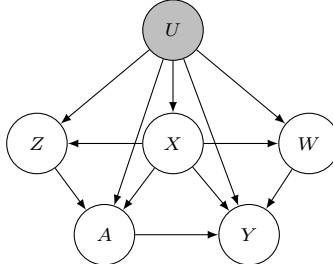

Figure 1: A causal DAG under the proximal causal inference framework.

We first consider two in-class optimal ITRs that map from $\mathcal{X} \times \mathcal{Z}$ to $\mathcal{A}$ and $\mathcal{X} \times \mathcal{W}$ to $\mathcal{A}$, respectively. To identify optimal ITRs that map from $\mathcal{X} \times \mathcal{Z}$ to $\mathcal{A}$, we make the following assumptions.

**Assumption 2.** *Completeness: For any $a \in \mathcal{A}, x \in \mathcal{X}$ and square-integrable function $g$, $\mathbb{E}[g(U) \mid Z, A = a, X = x] = 0$ almost surely if and only if $g(U) = 0$ almost surely.*

**Assumption 3.** *Existence of outcome confounding bridge: There exists an outcome confounding bridge function $h(w, a, x)$ that solves the following equation*

$$\mathbb{E}[Y|Z, A, X] = \mathbb{E}[h(W, A, X)|Z, A, X],$$

*almost surely.*

The completeness Assumption 2 is a technical condition central to the study of sufficiency in foundational theory of statistical inference. It essentially assumes that $Z$ has sufficient variability with respect to the variability of $U$. We refer to Tchetgen Tchetgen et al. (2020) and Miao et al. (2022) for further discussions regarding the completeness condition. Assumption 3 defines a so-called inverse problem known as a Fredholm integral equation of the first kind through an outcome confounding bridge. The technical conditions for the existence of a solution to a Fredholm integral equation can be found in Kress et al. (1989).

Let $\mathcal{D}_{\mathcal{Z}}$ be an ITR class that includes all measurable functions mapping from $\mathcal{X} \times \mathcal{Z}$ to $\mathcal{A}$. As shown in Qi et al. (2023), under Assumptions 1, 2 and 3, for any $d_z \in \mathcal{D}_{\mathcal{Z}}$, the value function $V(d_z)$ can be nonparametrically identified by

$$V(d_z) = \mathbb{E}[h(W, d_z(X, Z), X)]. \tag{1}$$

Furthermore, the in-class optimal treatment regime $d_z^* \in \arg\max_{d_z \in \mathcal{D}_{\mathcal{Z}}} V(d_z)$ is given by

$$d_z^*(X, Z) = \text{sign}\{\mathbb{E}[h(W, 1, X) - h(W, -1, X)|X, Z]\}.$$

On the other hand, to identify optimal ITRs that map from $\mathcal{X} \times \mathcal{W}$ to $\mathcal{A}$, we make the following assumptions.

**Assumption 4.** *Completeness: For any $a \in \mathcal{A}, x \in \mathcal{X}$ and square-integrable function $g$, $\mathbb{E}[g(U) \mid W, A = a, X = x] = 0$ almost surely if and only if $g(U) = 0$ almost surely.*

**Assumption 5.** *Existence of treatment confounding bridge: There exists a treatment confounding bridge function $q(z, a, x)$ that solves the following equation*

$$\frac{1}{\mathbb{P}(A = a|W, X)} = \mathbb{E}[q(Z, a, X)|W, A = a, X],$$

*almost surely.*

Similar to Assumptions 2 and 3, Assumption 4 assumes that $W$ has sufficient variability relative to the variability of $U$, and Assumption 5 defines another Fredholm integral equation of the first kind through a treatment confounding bridge $q$.

Let $\mathcal{D}_\mathcal{W}$ be another ITR class that includes all measurable functions mapping from $\mathcal{X} \times \mathcal{W}$ to $\mathcal{A}$. As shown in Qi et al. (2023), under Assumptions 1, 4 and 5, for any $d_w \in \mathcal{D}_\mathcal{W}$, the value function $V(d_w)$ can be nonparametrically identified by

$$V(d_w) = \mathbb{E}[Yq(Z, A, X)\mathbb{I}\{d_w(X, W) = A\}]. \tag{2}$$

The in-class optimal treatment regime $d_w^* \in \arg\max_{d_w \in \mathcal{D}_\mathcal{W}} V(d_w)$ is given by

$$d_w^*(X, W) = \text{sign}\{\mathbb{E}[Yq(Z, 1, X)\mathbb{I}\{A = 1\} - Yq(Z, -1, X)\mathbb{I}\{A = -1\}|X, W]\}.$$

Moreover, Qi et al. (2023) consider the ITR class $\mathcal{D}_\mathcal{Z} \cup \mathcal{D}_\mathcal{W}$ and propose a maximum proximal learning optimal regime based on this ITR class. For any $d_{z\cup w} \in \mathcal{D}_\mathcal{Z} \cup \mathcal{D}_\mathcal{W}$, under Assumptions 1-5, the value function $V(d_{z\cup w})$ for any $d_{z\cup w} \in \mathcal{D}_\mathcal{Z} \cup \mathcal{D}_\mathcal{W}$ can be identified by

$$V(d_{z\cup w}) = \mathbb{I}\{d_{z\cup w} \in \mathcal{D}_\mathcal{Z}\}\mathbb{E}[h(W, d_{z\cup w}(X, Z), X)] + \mathbb{I}\{d_{z\cup w} \in \mathcal{D}_\mathcal{W}\}\mathbb{E}[Yq(Z, A, X)\mathbb{I}\{d_{z\cup w}(X, W) = A\}]. \tag{3}$$

The optimal ITR within this class is given by $d_{z\cup w}^* \in \arg\max_{d_{z\cup w} \in \mathcal{D}_\mathcal{Z} \cup \mathcal{D}_\mathcal{W}} V(d_{z\cup w})$, and they show that the corresponding optimal value function takes the maximum value between two optimal in-class ITRs, i.e.,

$$V(d_{z\cup w}^*) = \max\{V(d_z^*), V(d_w^*)\}.$$

## 2.3 Optimal decision-making based on two confounding bridges

As discussed in the previous section, given that neither $\mathbb{E}[Y(a)|X, U]$ nor $\mathbb{E}[Y(a)|X, W, Z]$ for any $a \in \mathcal{A}$ may be identifiable under the proximal causal inference setting, one might nevertheless consider ITRs mapping from $\mathcal{X} \times \mathcal{W}$ to $\mathcal{A}$, from $\mathcal{X} \times \mathcal{Z}$ to $\mathcal{A}$, from $\mathcal{X} \times \mathcal{W} \times \mathcal{Z}$ to $\mathcal{A}$ as well as from $\mathcal{X}$ to $\mathcal{A}$. Intuitively, policy-makers might want to use as much information as they can to facilitate their decision-making. Therefore, ITRs mapping from $\mathcal{X} \times \mathcal{W} \times \mathcal{Z}$ to $\mathcal{A}$ are of great interest if information regarding $(X, W, Z)$ is available.

As a result, a natural question arises: is there an ITR mapping from $\mathcal{X} \times \mathcal{W} \times \mathcal{Z}$ to $\mathcal{A}$ which dominates existing ITRs proposed in the literature? In this section, we answer this question by proposing a novel optimal ITR and showing its superiority in terms of global welfare.

We first consider the following class of ITRs that map from $\mathcal{X} \times \mathcal{W} \times \mathcal{Z}$ to $\mathcal{A}$,

$$\mathcal{D}_{\mathcal{Z}\mathcal{W}}^\Pi \triangleq \{d_{zw}^\pi : d_{zw}^\pi(X, W, Z) = \pi(X)d_z(X, Z) + (1 - \pi(X))d_w(X, W), d_z \in \mathcal{D}_\mathcal{Z}, d_w \in \mathcal{D}_\mathcal{W}, \pi \in \Pi\},$$

where $\Pi$ is the policy class containing all measurable functions $\pi : \mathcal{X} \to \{0, 1\}$ that indicate the individualized predilection between $d_z$ and $d_w$.

**Remark 1.** *Note that $\mathcal{D}_\mathcal{Z}, \mathcal{D}_\mathcal{W}$ and $\mathcal{D}_\mathcal{Z} \cup \mathcal{D}_\mathcal{W}$ are subsets of $\mathcal{D}_{\mathcal{Z}\mathcal{W}}^\Pi$ with a particular choice of $\pi$. For example, $\mathcal{D}_\mathcal{Z}$ is $\mathcal{D}_{\mathcal{Z}\mathcal{W}}^\Pi$ with restriction on $\pi(X) = 1$; $\mathcal{D}_\mathcal{W}$ is $\mathcal{D}_{\mathcal{Z}\mathcal{W}}^\Pi$ with restriction on $\pi(X) = 0$; $\mathcal{D}_\mathcal{Z} \cup \mathcal{D}_\mathcal{W}$ is $\mathcal{D}_{\mathcal{Z}\mathcal{W}}^\Pi$ with restriction on $\pi(X) = 1$ or $\pi(X) = 0$.*

In the following theorem, we demonstrate that by leveraging the treatment and outcome confounding bridge functions, we can nonparametrically identify the value function over the policy class $\mathcal{D}_{\mathcal{Z}\mathcal{W}}^\Pi$, i.e., $V(d_{zw}^\pi)$ for $d_{zw}^\pi \in \mathcal{D}_{\mathcal{Z}\mathcal{W}}^\Pi$.

**Theorem 1.** *Under Assumptions 1-5, for any $d_{zw}^\pi \in \mathcal{D}_{\mathcal{Z}\mathcal{W}}^\Pi$, the value function $V(d_{zw}^\pi)$ can be nonparametrically identified by*

$$V(d_{zw}^\pi) = \mathbb{E}[\pi(X)h(W, d_z(X, Z), X) + (1 - \pi(X))Yq(Z, A, X)\mathbb{I}\{d_w(X, W) = A\}]. \tag{4}$$

One of the key ingredients of our constructed new policy class $\mathcal{D}_{\mathcal{Z}\mathcal{W}}^\Pi$ is the choice of $\pi(\cdot)$. It suggests an individualized strategy for treatment decisions between the two given treatment regimes. Because we are interested in policy learning, a suitable choice of $\pi(\cdot)$ that leads to a larger value function is more desirable. Therefore, we construct the following $\bar{\pi}(X; d_z, d_w)$,

$$\bar{\pi}(X; d_z, d_w) \triangleq \mathbb{I}\{\mathbb{E}[h(W, d_z(X, Z), X)|X] \geq \mathbb{E}[Yq(Z, A, X)\mathbb{I}\{d_w(X, W) = A\}|X]\}. \tag{5}$$

In addition, given any $d_z \in \mathcal{D}_\mathcal{Z}$ and $d_w \in \mathcal{D}_\mathcal{W}$, we define

$$d_{zw}^{\bar{\pi}}(X, W, Z) \triangleq \bar{\pi}(X; d_z, d_w)d_z(X, Z) + (1 - \bar{\pi}(X; d_z, d_w))d_w(X, W).$$

We then obtain the following result which justifies the superiority of $\bar{\pi}$.

**Theorem 2.** *Under Assumptions 1-5, for any $d_z \in \mathcal{D}_\mathcal{Z}$ and $d_w \in \mathcal{D}_\mathcal{W}$,*

$$V(d_{zw}^{\bar{\pi}}) \geq \max\{V(d_z), V(d_w)\}.$$

Theorem 2 establishes that for the particular choice of $\bar{\pi}$ given in (5), the value function of $d_{zw}^{\bar{\pi}}$ is no smaller than that of $d_z$ and $d_w$ for any $d_z \in \mathcal{D}_\mathcal{Z}$, and $d_w \in \mathcal{D}_\mathcal{W}$. Consequently, Theorem 2 holds for $d_z^*$ and $d_w^*$. Hence, we propose the following optimal ITR $d_{zw}^{\bar{\pi}*}$,

$$d_{zw}^{\bar{\pi}*}(X, W, Z) \triangleq \bar{\pi}(X; d_z^*, d_w^*)d_z^*(X, Z) + (1 - \bar{\pi}(X; d_z^*, d_w^*))d_w^*(X, W),$$

and we have the following corollary.

**Corollary 1.** *Under Assumptions 1-5, we have that*

$$V(d_{zw}^{\bar{\pi}*}) \geq \max\{V(d_z^*), V(d_w^*), V(d_{z \cup w}^*)\}.$$

Corollary 1 essentially states that the value of $d_{zw}^{\bar{\pi}*}$ dominates that of $d_z^*$, $d_w^*$, as well as $d_{z \cup w}^*$. Moreover, the proposition below demonstrates the optimality of $d_{zw}^{\bar{\pi}*}$ within the proposed class.

**Proposition 1.** *Under Assumptions 1-5, we have that*

$$d_{zw}^{\bar{\pi}*} \in \arg \max_{d_{zw}^\pi \in \mathcal{D}_{\mathcal{Z}\mathcal{W}}^\Pi} V(d_{zw}^\pi).$$

Therefore, $d_{zw}^{\bar{\pi}*}$ is an optimal ITR of policymakers' interest.

# 3 Statistical Learning and Optimization

## 3.1 Estimation of the optimal ITR $d_{zw}^{\bar{\pi}*}$

The estimation of $d_{zw}^{\bar{\pi}*}$ consists of four steps: (i) estimation of confounding bridges $h$ and $q$; (ii) estimation of preliminary ITRs $d_z^*$ and $d_w^*$; (iii) estimation of $\bar{\pi}(X; d_z^*, d_w^*)$; and (iv) learning $d_{zw}^{\bar{\pi}*}$ based on (ii) and (iii). The estimation problem (i) has been developed by Cui et al. (2023); Miao et al. (2018b) using the generalized method of moments, Ghassami et al. (2022); Kallus et al. (2021) by a min-max estimation (Dikkala et al., 2020) using kernels, and Kompa et al. (2022) using deep learning; and (ii) has been developed by Qi et al. (2023). We restate estimation of (i) and (ii) for completeness. With regard to (i), recall that Assumptions 3 and 5 imply the following conditional moment restrictions

$$\mathbb{E}[Y - h(W, A, X)|Z, A, X] = 0,$$
$$\mathbb{E}[1 - \mathbb{I}\{A = a\}q(Z, a, X)|W, X] = 0, \forall a \in \mathcal{A}.$$

respectively. Kompa et al. (2022) propose a deep neural network approach to estimating bridge functions which avoids the reliance on kernel methods. We adopt this approach in our simulation and details can be found in the Appendix.

To estimate $d_z^*$, we consider classification-based approaches according to Zhang et al. (2012); Zhao et al. (2012). Under Assumptions 1, 2 and 3, maximizing the value function in (1) is equivalent to minimizing the following classification error

$$\mathbb{E}[\{h(W, 1, X) - h(W, -1, X)\}\mathbb{I}\{d_z(X, Z) \neq 1\}] \tag{6}$$

over $d_z \in \mathcal{D}_\mathcal{Z}$. By choosing some measurable decision function $g_z \in \mathcal{G}_\mathcal{Z} : \mathcal{X} \times \mathcal{Z} \to \mathbb{R}$, we let $d_z(X, Z) = \text{sign}(g_z(X, Z))$. We consider the following empirical version of (6),

$$\min_{g_z \in \mathcal{G}_z} \mathbb{P}_n[\{\hat{h}(W, 1, X) - \hat{h}(W, -1, X)\}\mathbb{I}\{g_z(X, Z) < 0\}].$$

Due to the non-convexity and non-smoothness of the sign operator, we replace the sign operator with a smooth surrogate function and adopt the hinge loss $\phi(x) = \max\{1 - x, 0\}$. By adding a penalty term $\rho_z \|g_z\|_{\mathcal{G}_{\mathcal{Z}}}^2$ to avoid overfitting, we solve

$$\hat{g}_z \in \arg\min_{g_z \in \mathcal{G}_z} \mathbb{P}_n[\{\hat{h}(W, 1, X) - \hat{h}(W, -1, X)\}\phi(g_z(X, Z))] + \rho_z \|g_z\|_{\mathcal{G}_{\mathcal{Z}}}^2, \quad (7)$$

where $\rho_z > 0$ is a tuning parameter. The estimated ITR then follows $\hat{d}_z(X, Z) = \text{sign}(\hat{g}_z(X, Z))$. Similarly, under Assumptions 1, 4 and 5, maximizing the value function in (2) is equivalent to minimizing the following classification error

$$\mathbb{E}[\{Yq(Z, 1, X)\mathbb{I}\{A = 1\} - Yq(Z, -1, X)\mathbb{I}\{A = -1\}\}\mathbb{I}\{d_w(X, W) \neq 1\}]$$

over $d_w \in \mathcal{D}_{\mathcal{W}}$. By the same token, the problem is transformed into minimizing the following empirical error

$$\hat{g}_w \in \arg\min_{g_w \in \mathcal{G}_{\mathcal{W}}} \mathbb{P}_n[\{Y\hat{q}(Z, 1, X)\mathbb{I}\{A = 1\} - Y\hat{q}(Z, -1, X)\mathbb{I}\{A = -1\}\}\phi(g_w(X, W))] + \rho_w \|g_w\|_{\mathcal{G}_{\mathcal{W}}}^2. \quad (8)$$

The estimated ITR is obtained via $\hat{d}_w(X, W) = \text{sign}(\hat{g}_w(X, W))$.

For problem (iii), given two preliminary ITRs, we construct an estimator $\hat{\pi}(X; \hat{d}_z, \hat{d}_w)$, that is, for $x \in \mathcal{X}$,

$$\hat{\pi}(x; \hat{d}_z, \hat{d}_w) = \mathbb{I}\{\hat{\delta}(x; \hat{d}_z, \hat{d}_w) \geq 0\},$$

where $\hat{\delta}(x; \hat{d}_z, \hat{d}_w)$ denotes a generic estimator of

$$\delta(x; \hat{d}_z, \hat{d}_w) \triangleq \mathbb{E}[h(W, \hat{d}_z(X, Z), X) - Yq(Z, A, X)\mathbb{I}\{\hat{d}_w(X, W) = A\}|X = x],$$

where the expectation is taken with respect to everything except $\hat{d}_z$ and $\hat{d}_w$. For example, the Nadaraya-Watson kernel regression estimator (Nadaraya, 1964) can be used, i.e., $\hat{\delta}(x; \hat{d}_z, \hat{d}_w)$ is expressed as

$$\frac{\sum_{i=1}^n \{\hat{h}(W_i, \hat{d}_z(x, Z_i), x) - Y_i\hat{q}(Z_i, A_i, x)\mathbb{I}\{\hat{d}_w(x, W_i) = A_i\}\}K(\frac{\|x - X_i\|_2}{\gamma})}{\sum_{i=1}^n K(\frac{\|x - X_i\|_2}{\gamma})},$$

where $K : \mathbb{R} \to \mathbb{R}$ is a kernel function such as Gaussian kernel, $\|\cdot\|_2$ denotes the $L_2$-norm, and $\gamma$ denotes the bandwidth.

Finally, given $\hat{d}_z, \hat{d}_w$ and $\hat{\pi}(X; \hat{d}_z, \hat{d}_w)$, $\hat{d}_{zw}^{\hat{\pi}}$ is estimated by the following plug-in regime,

$$\hat{d}_{zw}^{\hat{\pi}}(X, W, Z) = \hat{\pi}(X; \hat{d}_z, \hat{d}_w)\hat{d}_z(X, Z) + (1 - \hat{\pi}(X; \hat{d}_z, \hat{d}_w))\hat{d}_w(X, W). \quad (9)$$

## 3.2 Theoretical guarantees for $\hat{d}_{zw}^{\hat{\pi}}$

In this subsection, we first present an optimality guarantee for the estimated ITR $\hat{d}_{zw}^{\hat{\pi}}$ in terms of its value function

$$V(\hat{d}_{zw}^{\hat{\pi}}) = \mathbb{E}[\hat{\pi}(X; \hat{d}_z, \hat{d}_w)h(W, \hat{d}_z(X, Z), X) + (1 - \hat{\pi}(X; \hat{d}_z, \hat{d}_w))Yq(Z, A, X)\mathbb{I}\{\hat{d}_w(X, W) = A\}],$$

where the expectation is taken with respect to everything except $\hat{\pi}, \hat{d}_z$ and $\hat{d}_w$.

We define an oracle optimal ITR which assumes $\bar{\pi}(X; \hat{d}_z, \hat{d}_w)$ is known,

$$\hat{d}_{zw}^{\bar{\pi}}(X, W, Z) \triangleq \bar{\pi}(X; \hat{d}_z, \hat{d}_w)\hat{d}_z(X, Z) + (1 - \bar{\pi}(X; \hat{d}_z, \hat{d}_w))\hat{d}_w(X, W).$$

The corresponding value function of this oracle optimal ITR is given by

$$V(\hat{d}_{zw}^{\bar{\pi}}) = \mathbb{E}[\bar{\pi}(X; \hat{d}_z, \hat{d}_w)h(W, \hat{d}_z(X, Z), X) + (1 - \bar{\pi}(X; \hat{d}_z, \hat{d}_w))Yq(Z, A, X)\mathbb{I}\{\hat{d}_w(X, W) = A\}],$$

where the expectation is taken with respect to everything except $\hat{d}_z$ and $\hat{d}_w$.

Then the approximation error incurred by estimating $\hat{\pi}(X; \hat{d}_z, \hat{d}_w)$ is given by

$$\mathbb{K}(\hat{\pi}) \stackrel{\triangle}{=} V(\hat{d}_{zw}^{\bar{\pi}}) - V(\hat{d}_{zw}^{\hat{\pi}}).$$

Moreover, we define the following gain

$$\mathbb{G}(\bar{\pi}) \stackrel{\triangle}{=} \min\{V(\hat{d}_{zw}^{\bar{\pi}}) - V(\hat{d}_z), V(\hat{d}_{zw}^{\bar{\pi}}) - V(\hat{d}_w)\}.$$

It is clear that this gain $\mathbb{G}(\bar{\pi})$ by introducing $\bar{\pi}$ is always non-negative as indicated by Theorem 2. Then we have the following excess value bound for the value of $\hat{d}_{zw}^{\hat{\pi}}$ compared to existing ones in the literature.

**Proposition 2.** *Under Assumptions 1-5,*

$$V(\hat{d}_{zw}^{\hat{\pi}}) = \max\{V(\hat{d}_z), V(\hat{d}_w)\} - \mathbb{K}(\hat{\pi}) + \mathbb{G}(\bar{\pi}) = V(\hat{d}_{z \cup w}) - \mathbb{K}(\hat{\pi}) + \mathbb{G}(\bar{\pi}).$$

Proposition 2 establishes a link between the value function of the estimated ITR $\hat{d}_{zw}^{\hat{\pi}}$, and that of $\hat{d}_z$, $\hat{d}_w$, and $\hat{d}_{z \cup w}$. As shown in Appendix G, $\mathbb{K}(\hat{\pi})$ diminishes as the sample size increases, therefore, $\hat{d}_{zw}^{\hat{\pi}}$ has a significant improvement compared to other optimal ITRs depending on the magnitude of $\mathbb{G}(\bar{\pi})$.

Furthermore, we establish the consistency of the proposed regime based on the following assumption, which holds for example when $\hat{d}_z$ and $\hat{d}_w$ are estimated using indirect methods.

**Assumption 6.** *For $\hat{d}_z, \hat{d}_w$, $E[h(W, \hat{d}_z(X, Z), X)|X] - E[h(W, d_z^*(X, Z), X)|X] = o_p(n^{-\xi})$ almost surely and $E[Yq(Z, A, X)\mathbb{I}\{\hat{d}_w(X, W) = A\}|X] - E[Yq(Z, A, X)\mathbb{I}\{d_w^*(X, W) = A\}|X] = o_p(n^{-\varphi})$ almost surely.*

**Proposition 3.** *Under Assumptions 1-6, we have $V(\hat{d}_{zw}^{\hat{\pi}}) \xrightarrow{p} V(d_{zw}^{\bar{\pi}*})$.*

## 4 Numerical Experiments

The data generating mechanism for $(X, A, Z, W, U)$ follows the setup proposed in Cui et al. (2023) and is summarized in Appendix I. To evaluate the performance of the proposed framework, we vary $b_1(X)$, $b_2(X)$, $b_3(X)$, $b_a$ and $b_w$ in $\mathbb{E}[Y|X, A, Z, W, U]$ to incorporate heterogeneous treatment effects including the settings considered in Qi et al. (2023). The adopted data generating mechanism is compatible with the following $h$ and $q$,

$$h(W, A, X) = b_0 + \{b_1(X) + b_a W + b_3(X)W\}\frac{1+A}{2} + b_w W + b_2(X)X,$$

$$q(Z, A, X) = 1 + \exp\left\{At_0 + At_z Z + t_a \frac{1+A}{2} + At_x X\right\},$$

where $t_0 = 0.25, t_z = -0.5, t_a = -0.125$, and $t_x = (0.25, 0.25)^T$. We derive preliminary optimal ITRs $d_z^*$ and $d_w^*$ in Appendix J, from which we can see that $X, Z, W$ are relevant variables for individualized decision-making.

We consider six scenarios in total, and the setups of varying parameters are deferred to Appendix I. For each scenario, training datasets $\{Y_i, A_i, X_i, Z_i, W_i\}_{i=1}^n$ are generated following the above mechanism with a sample size $n = 1000$. For each training dataset, we then apply the aforementioned methods to learn the optimal ITR. In particular, the preliminary ITRs $\hat{d}_z$ and $\hat{d}_w$ are estimated using a linear decision rule, and $\hat{\pi}(x; \hat{d}_z, \hat{d}_w)$ is estimated using a Gaussian kernel. More details can be found in the Appendix K.

To evaluate the estimated treatment regimes, we consider the following generating mechanism for testing datasets: $X \sim \mathcal{N}(\Gamma_x, \Sigma_x)$,

$$(Z, W, U)|X \sim \mathcal{N}\left\{\begin{pmatrix} \alpha_0 + \alpha_a p_a + \alpha_x X \\ \mu_0 + \mu_a p_a + \mu_x X \\ \kappa_0 + \kappa_a p_a + \kappa_x X \end{pmatrix}, \quad \Sigma = \begin{pmatrix} \sigma_z^2 & \sigma_{zw} & \sigma_{zu} \\ \sigma_{zw} & \sigma_w^2 & \sigma_{wu} \\ \sigma_{zu} & \sigma_{wu} & \sigma_u^2 \end{pmatrix}\right\},$$

where the parameter settings can be found in Appendix I. The testing dataset is generated with a size 10000, and the empirical value function for the estimated ITR is used as a performance measure. The simulations are replicated 200 times. To validate our approach and demonstrate its superiority, we have also computed empirical values for other optimal policies, including existing optimal ITRs for proximal causal inference, as discussed in Section 2.2, along with optimal ITRs generated through causal forest (Athey and Wager, 2019) and outcome weighted learning (Zhao et al., 2012).

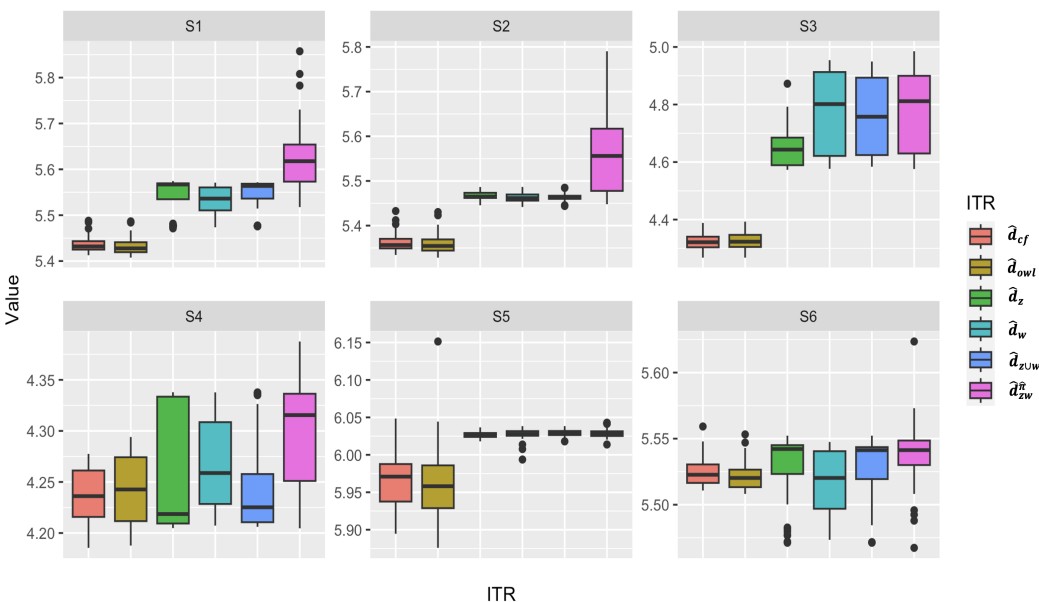

Figure 2: Boxplots of the empirical value functions ($\hat{d}_{cf}$ and $\hat{d}_{owl}$ denote estimated ITRs using causal forest and outcome weighted learning respectively).

Figure 2 presents the empirical value functions of different optimal ITRs for the six scenarios. As expected, $\hat{d}_z, \hat{d}_w, \hat{d}_{z\cup w}$, and $\hat{d}_{zw}^{\hat{\pi}}$ consistently outperform $\hat{d}_{cf}$ and $\hat{d}_{owl}$, which highlights their effectiveness in addressing unmeasured confounding. Meanwhile, across all scenarios, $\hat{d}_{zw}^{\hat{\pi}}$ yields superior or comparable performance compared to the other estimated treatment regimes, which justifies the statements made in Sections 2 and 3. In addition, as can be seen in Scenario 5, all ITRs relying on the proximal causal inference framework perform similarly, which is not surprising as $\hat{d}_z(X, Z)$ and $\hat{d}_w(X, W)$ agree for most subjects. To further underscore the robust performance of our proposed approach, we include additional results with a changed sample size and a modified behavior policy in Appendix L.

## 5    Real Data Application

In this section, we demonstrate the proposed optimal ITR via a real dataset originally designed to measure the effectiveness of right heart catheterization (RHC) for ill patients in intensive care units (ICU), under the Study to Understand Prognoses and Preferences for Outcomes and Risks of Treatments (SUPPORT, Connors et al. (1996)). These data have been re-analyzed in a number of papers in both causal inference and survival analysis literature with assuming unconfoundednss (Cui and Tchetgen Tchetgen, 2023; Tan, 2006, 2020, 2019; Vermeulen and Vansteelandt, 2015) or accounting for unmeasured confounding (Cui et al., 2023; Lin et al., 1998; Qi et al., 2023; Tchetgen Tchetgen et al., 2020; Ying et al., 2022).

There are 5735 subjects included in the dataset, in which 2184 were treated (with $A = 1$) and 3551 were untreated (with $A = -1$). The outcome $Y$ is the duration from admission to death or censoring. Overall, 3817 patients survived and 1918 died within 30 days. Following Tchetgen Tchetgen et al. (2020), we collect 71 covariates including demographic factors, diagnostic information, estimated survival probability, comorbidity, vital signs, physiological status, and functional status (see

Hirano and Imbens (2001) for additional discussion on covariates). Confounding in this study stems from the fact that ten physiological status measures obtained from blood tests conducted at the initial phase of admission may be susceptible to significant measurement errors. Furthermore, besides the lab measurement errors, whether other unmeasured confounding factors exist is unknown to the data analyst. Because variables measured from these tests offer only a single snapshot of the underlying physiological condition, they have the potential to act as confounding proxies. We consider a total of four settings, varying the number of selected proxies from 4 to 10. Within each setting, treatment-inducing proxies are first selected based on their strength of association with the treatment (determined through logistic regression of $A$ on $L$), and outcome-inducing proxies are then chosen based on their association with the outcome (determined through linear regression of $Y$ on $A$ and $L$). Excluding the selected proxy variables, other measured covariates are included in $X$. We then estimate $\hat{d}_z, \hat{d}_w, \hat{d}_{z \cup w}$, and $\hat{d}_{zw}^{\hat{\pi}}$ using the SUPPORT dataset in a manner similar to that described in Section 4, with the goal optimizing the patients' 30-day survival after their entrance into the ICU.

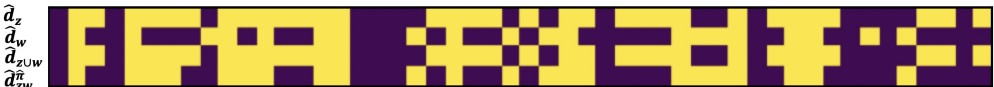

Figure 3: Graphical representation of concordance between estimated ITRs.

The estimated value functions of our proposed ITR, alongside existing ones, are summarized in Appendix M. As can be seen, our proposed regime has the largest value among all settings. For a visual representation of the concordance between the estimated optimal ITRs, we refer to Figure 3 (results from Setting 1). The horizontal ordinate represents the 50 selected subjects and the vertical axis denotes the decisions made from corresponding ITRs. The purple and yellow blocks stand for being recommended treatment values of -1 and 1 respectively. For the subjects with purple or yellow columns, $\hat{d}_z(X, Z) = \hat{d}_w(X, W)$, which leads to the same treatment decision for the other two ITRs. For columns with mixed colors, $\hat{d}_z(X, Z)$ and $\hat{d}_w(X, W)$ disagree. We see that in this case $\hat{d}_{z \cup w}(X, W, Z)$ always agree with $\hat{d}_z(X, Z)$, while $\hat{d}_{zw}^{\hat{\pi}}(X, W, Z)$ take values from $\hat{d}_z(X, Z)$ or $\hat{d}_w(X, W)$ depending on the individual criteria of the subjects as indicated by $\hat{\pi}$. In addition to the quantitative analysis, we have also conducted a qualitative assessment of the estimated regime to validate its performance. For further details, please refer to Appendix M.

## 6    Discussion

We acknowledge several limitations of our work. Firstly, the proximal causal inference framework relies on the validity of treatment- and outcome-inducing confounding proxies. When the assumptions are violated, the proximal causal inference estimators can be biased even if unconfoundedness on the basis of measured covariates in fact holds. Therefore, one needs to carefully sort out proxies especially when domain knowledge is lacking. Secondly, while the proposed regime significantly improves upon existing methods both theoretically and numerically, it is not yet shown to be the sharpest under our considered model. It is still an open question to figure out if a more general policy class could be considered. Thirdly, our established theory provides consistency and superiority of our estimated regime. It is of great interest to derive convergence rates for Propositions 2 and 3 following Jiang (2017). In addition, it may be challenging to develop inference results for the value function of the estimated optimal treatment regimes, and further studies are warranted.

## Acknowledgments and Disclosure of Funding

Yifan Cui was supported by the National Natural Science Foundation of China.

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
