# Supplementary Material

## A   Proof of identification (3)

The proof is straightforward. We state it here for clarity and completeness. Note that

$\mathbb{I}\{d_{z\cup w}(X, W, Z) = 1\} = \mathbb{I}\{d_{z\cup w} \in \mathcal{D}_{\mathcal{Z}}\}\mathbb{I}\{d_{z\cup w}(X, Z) = 1\} + \mathbb{I}\{d_{z\cup w} \in \mathcal{D}_{\mathcal{W}}\}\mathbb{I}\{d_{z\cup w}(X, W) = 1\}$,

$\mathbb{I}\{d_{z\cup w}(X, W, Z) = -1\} = \mathbb{I}\{d_{z\cup w} \in \mathcal{D}_{\mathcal{Z}}\}\mathbb{I}\{d_{z\cup w}(X, Z) = -1\} + \mathbb{I}\{d_{z\cup w} \in \mathcal{D}_{\mathcal{W}}\}\mathbb{I}\{d_{z\cup w}(X, W) = -1\}$.

Therefore, we have

$$
\begin{aligned}
\mathbb{E}[Y(1)\mathbb{I}\{d_{z\cup w}(X, W, Z) = 1\}] &= \mathbb{E}[Y(1)\mathbb{I}\{d_{z\cup w} \in \mathcal{D}_{\mathcal{Z}}\}\mathbb{I}\{d_{z\cup w}(X, Z) = 1\} \\
&\quad + Y(1)\mathbb{I}\{d_{z\cup w} \in \mathcal{D}_{\mathcal{W}}\}\mathbb{I}\{d_{z\cup w}(X, W) = 1\}] \\
&= \mathbb{I}\{d_{z\cup w} \in \mathcal{D}_{\mathcal{Z}}\}\mathbb{E}[Y(1)\mathbb{I}\{d_{z\cup w}(X, Z) = 1\}] \\
&\quad + \mathbb{I}\{d_{z\cup w} \in \mathcal{D}_{\mathcal{W}}\}\,\mathbb{E}[Y(1)\mathbb{I}\{d_{z\cup w}(X, W) = 1\}].
\end{aligned}
$$

Similarly,

$$
\begin{aligned}
\mathbb{E}[Y(-1)\mathbb{I}\{d_{z\cup w}(X, W, Z) = -1\}] &= \mathbb{I}\{d_{z\cup w} \in \mathcal{D}_{\mathcal{Z}}\}\mathbb{E}[Y(-1)\mathbb{I}\{d_{z\cup w}(X, Z) = -1\}] \\
&\quad + \mathbb{I}\{d_{z\cup w} \in \mathcal{D}_{\mathcal{W}}\}\,\mathbb{E}[Y(-1)\mathbb{I}\{d_{z\cup w}(X, W) = -1\}].
\end{aligned}
$$

So

$$
\begin{aligned}
V(d_{z\cup w}) &= \mathbb{E}[Y(1)\mathbb{I}\{d_{z\cup w}(X, W, Z) = 1\}] + \mathbb{E}[Y(-1)\mathbb{I}\{d_{z\cup w}(X, W, Z) = -1\}] \\
&= \mathbb{I}\{d_{z\cup w} \in \mathcal{D}_{\mathcal{Z}}\}\mathbb{E}[Y(1)\mathbb{I}\{d_{z\cup w}(X, Z) = 1\} + Y(-1)\mathbb{I}\{d_{z\cup w}(X, Z) = -1\}] \\
&\quad + \mathbb{I}\{d_{z\cup w} \in \mathcal{D}_{\mathcal{W}}\}\mathbb{E}[Y(1)\mathbb{I}\{d_{z\cup w}(X, W) = 1\} + Y(-1)\mathbb{I}\{d_{z\cup w}(X, W) = -1\}] \\
&= \mathbb{I}\{d_{z\cup w} \in \mathcal{D}_{\mathcal{Z}}\}\mathbb{E}[h(W, d_{z\cup w}(X, Z), X)] + \mathbb{I}\{d_{z\cup w} \in \mathcal{D}_{\mathcal{W}}\}\mathbb{E}[Y q(Z, A, X)\mathbb{I}\{d_{z\cup w}(X, W) = A\}],
\end{aligned}
$$

where the last equality holds due to identification results (1) and (2).

## B   Proof of Theorem 1

Recall that $V(d_{zw}^{\pi}) = \mathbb{E}[Y(1)\mathbb{I}\{d_{zw}^{\pi}(X, W, Z) = 1\} + Y(-1)\mathbb{I}\{d_{zw}^{\pi}(X, W, Z) = -1\}]$, we essentially need to consider the first term $\mathbb{E}[Y(1)\mathbb{I}\{d_{zw}^{\pi}(X, W, Z) = 1\}]$. Note that

$$\mathbb{I}\{d_{zw}^{\pi}(X, W, Z) = 1\} = \mathbb{I}\{\pi(X) = 1\}\mathbb{I}\{d_z(X, Z) = 1\} + \mathbb{I}\{\pi(X) = 0\}\mathbb{I}\{d_w(X, W) = 1\},$$

we have

$$
\begin{aligned}
\mathbb{E}[Y(1)\mathbb{I}\{d_{zw}^{\pi}(X, W, Z) = 1\}] &= \mathbb{E}[Y(1)\mathbb{I}\{\pi(X) = 1\}\mathbb{I}\{d_z(X, Z) = 1\} + Y(1)\mathbb{I}\{\pi(X) = 0\}\mathbb{I}\{d_w(X, W) = 1\}] \\
&= \mathbb{E}[\mathbb{I}\{\pi(X) = 1\}\mathbb{E}[Y(1)\mathbb{I}\{d_z(X, Z) = 1\}|X] \\
&\quad + \mathbb{I}\{\pi(X) = 0\}\mathbb{E}[Y(1)\mathbb{I}\{d_w(X, W) = 1\}|X]].
\end{aligned}
$$

By leveraging the outcome confounding bridge, we have

$$
\begin{aligned}
\mathbb{E}[Y(1)\mathbb{I}\{d_z(X, Z) = 1\}|X] &= \mathbb{E}[\mathbb{E}[Y(1)|X, Z]\mathbb{I}\{d_z(X, Z) = 1\}|X] \\
&= \mathbb{E}[\mathbb{E}[\mathbb{E}[Y(1)|X, Z, U]|X, Z]\mathbb{I}\{d_z(X, Z) = 1\}|X] \\
&= \mathbb{E}[\mathbb{E}[\mathbb{E}[Y|X, U, A = 1]|X, Z]\mathbb{I}\{d_z(X, Z) = 1\}|X] \\
&= \mathbb{E}[\mathbb{E}[\mathbb{E}[h(W, 1, X)|X, U]|X, Z]\mathbb{I}\{d_z(X, Z) = 1\}|X] \\
&= \mathbb{E}[\mathbb{E}[\mathbb{E}[h(W, 1, X)|X, Z, U]|X, Z]\mathbb{I}\{d_z(X, Z) = 1\}|X] \\
&= \mathbb{E}[h(W, 1, X)\mathbb{I}\{d_z(X, Z) = 1\}|X],
\end{aligned}
$$

where the third equality is due to Assumption 1, the fourth equality can be verified by Theorem 1 in Miao et al. (2018a) under Assumptions 2 and 3, and the fifth equality is due to Assumption 1. Moreover, by leveraging the treatment confounding bridge, we have

$$
\begin{aligned}
\mathbb{E}[Y(1)\mathbb{I}\{d_w(X, W) = 1\}|X] &= \mathbb{E}[\mathbb{E}[Y(1)|X, W]\mathbb{I}\{d_w(X, W) = 1\}|X] \\
&= \mathbb{E}[\mathbb{E}[\mathbb{E}[Y(1)|X, W, U]|X, W]\mathbb{I}\{d_w(X, W) = 1\}|X] \\
&= \mathbb{E}[\mathbb{E}[\mathbb{E}[Y(1)|X, W, U, A = 1]|X, W]\mathbb{I}\{d_w(X, W) = 1\}|X] \\
&= \mathbb{E}[\mathbb{E}[\mathbb{E}[Y(1)|X, W, U, A = 1]\mathbb{E}[q(Z, 1, X)|X, U, A = 1] \\
&\quad \mathbb{P}(A = 1|X, U)|X, W]\mathbb{I}\{d_w(X, W) = 1\}|X] \\
&= \mathbb{E}[\mathbb{E}[\mathbb{E}[Y q(Z, 1, X)\mathbb{I}\{A = 1\}|X, U, W]|X, W]\mathbb{I}\{d_w(X, W) = 1\}|X] \\
&= \mathbb{E}[Y q(Z, 1, X)\mathbb{I}\{A = 1\}\mathbb{I}\{d_w(X, W) = 1\}|X],
\end{aligned}
$$

where the third equality is due to Assumption 1, the fourth equality is implied by Theorem 2.2 of Cui et al. (2023) under Assumptions 4 and 5, and the fifth equality is due to Assumption 1. Therefore,

$$
\begin{aligned}
\mathbb{E}[Y(1)\mathbb{I}\{d_{zw}^{\pi}(X,W,Z)=1\}] &= \mathbb{E}[\mathbb{I}\{\pi(X)=1\}\mathbb{E}[Y(1)\mathbb{I}\{d_z(X,Z)=1\}|X] \\
&\quad + \mathbb{I}\{\pi(X)=0\}\mathbb{E}[Y(1)\mathbb{I}\{d_w(X,W)=1\}|X]] \\
&= \mathbb{E}[\mathbb{I}\{\pi(X)=1\}\mathbb{E}[h(W,1,X)\mathbb{I}\{d_z(X,Z)=1\}|X] \\
&\quad + \mathbb{I}\{\pi(X)=0\}\mathbb{E}[Yq(Z,1,X)\mathbb{I}\{A=1\}\mathbb{I}\{d_w(X,W)=1\}|X]] \\
&= \mathbb{E}[\mathbb{I}\{\pi(X)=1\}h(W,1,X)\mathbb{I}\{d_z(X,Z)=1\} \\
&\quad + \mathbb{I}\{\pi(X)=0\}Yq(Z,1,X)\mathbb{I}\{A=1\}\mathbb{I}\{d_w(X,W)=1\}]. \quad (10)
\end{aligned}
$$

Similarly, as

$$
\mathbb{I}\{d_{zw}^{\pi}(X,W,Z)=-1\} = \mathbb{I}\{\pi(X)=1\}\mathbb{I}\{d_z(X,Z)=-1\} + \mathbb{I}\{\pi(X)=0\}\mathbb{I}\{d_w(X,W)=-1\},
$$

we have

$$
\begin{aligned}
&\mathbb{E}[Y(-1)\mathbb{I}\{d_{zw}^{\pi}(X,W,Z)=-1\}] \\
&= \mathbb{E}[\mathbb{I}\{\pi(X)=1\}h(W,-1,X)\mathbb{I}\{d_z(X,Z)=-1\} \\
&\quad + \mathbb{I}\{\pi(X)=0\}Yq(Z,-1,X)\mathbb{I}\{A=-1\}\mathbb{I}\{d_w(X,W)=-1\}]. \quad (11)
\end{aligned}
$$

Combining (10) and (11), we have

$$
\begin{aligned}
V(d_{zw}^{\pi}) &= \mathbb{E}[Y(1)\mathbb{I}\{d_{zw}^{\pi}(X,W,Z)=1\} + Y(-1)\mathbb{I}\{d_{zw}^{\pi}(X,W,Z)=-1\}] \\

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

}^{\bar{\pi}} \in \mathcal{D}_{\mathcal{ZW}}^{\bar{\pi}}} V(d_{zw}^{\bar{\pi}}),$$

where $\mathcal{D}_{\mathcal{ZW}}^{\bar{\pi}} \triangleq \{d_{zw}^{\bar{\pi}} : d_{zw}^{\bar{\pi}}(X,W,Z) = \bar{\pi}(X)d_z(X,Z) + (1 - \bar{\pi}(X))d_w(X,W), d_z \in \mathcal{D}_{\mathcal{Z}}, d_w \in \mathcal{D}_{\mathcal{W}}\}$. Recall that
$$
\begin{aligned}
d_z^*(X,Z) &= \text{sign}\{\mathbb{E}[h(W, 1, X) - h(W, -1, X)|X, Z]\}, \\
d_w^*(X,W) &= \text{sign}\{\mathbb{E}[Yq(Z,A,X)\mathbb{I}\{A = 1\} - Yq(Z,A,X)\mathbb{I}\{A = -1\}|X, W]\},
\end{aligned}
$$

we have
$$
\begin{aligned}
\mathbb{E}[h(W, d_z^*(X,Z), X)|X, Z] &\geq \mathbb{E}[h(W, d_z(X,Z), X)|X, Z], \\
\mathbb{E}[Yq(Z,A,X)\mathbb{I}\{A = d_w^*(X,W)\}|X, W] &\geq \mathbb{E}[Yq(Z,A,X)\mathbb{I}\{A = d_w(X,W)\}|X, W].
\end{aligned}
$$

Taking expectation with respect to $Z$ and $W$ given $X$ respectively, we have
$$\mathbb{E}[\mathbb{E}[h(W, d_z^*(X,Z), X)|X, Z]|X] \geq \mathbb{E}[\mathbb{E}[h(W, d_z(X,Z), X)|X, Z]|X], \tag{12}$$
$$\mathbb{E}[\mathbb{E}[Yq(Z,A,X)\mathbb{I}\{d_w^*(X,W) = A\}|X, W]|X] \geq \mathbb{E}[\mathbb{E}[Yq(Z,A,X)\mathbb{I}\{d_w(X,W) = A\}|X, W]|X]. \tag{13}$$

By the proof given in Section C, we have that
$$V(d_{zw}^{\bar{\pi}*}) = \mathbb{E}[\max\{\mathbb{E}[h(W, d_z^*(X,Z), X)|X], \mathbb{E}[Yq(Z,A,X)\mathbb{I}\{d_w^*(X,W) = A\}|X]\}],$$

$$V(d_{zw}^{\bar{\pi}}) = \mathbb{E}[\max\{\mathbb{E}[h(W, d_z(X,Z), X)|X], \mathbb{E}[Yq(Z,A,X)\mathbb{I}\{d_w(X,W) = A\}|X]\}].$$

Therefore,
$$V(d_{zw}^{\bar{\pi}*}) = \mathbb{E}[\max\{\mathbb{E}[\mathbb{E}[h(W, d_z^*(X,Z), X)|X, Z]|X], \mathbb{E}[\mathbb{E}[Yq(Z,A,X)\mathbb{I}\{d_w^*(X,W) = A\}|X, W]|X]\}],$$

$$V(d_{zw}^{\bar{\pi}}) = \mathbb{E}[\max\{\mathbb{E}[\mathbb{E}[h(W, d_z(X,Z), X)|X, Z]|X], \mathbb{E}[\mathbb{E}[Yq(Z,A,X)\mathbb{I}\{d_w(X,W) = A\}|X, W]|X]\}].$$

From (12) and (13), we have $V(d_{zw}^{\bar{\pi}*}) \geq V(d_{zw}^{\bar{\pi}})$ for any $d_{zw}^{\bar{\pi}} \in \mathcal{D}_{\mathcal{ZW}}^{\bar{\pi}}$, which implies that $d_{zw}^{\bar{\pi}*}$ is the maximizer of $V(d_{zw}^{\pi})$.

# F  Proof of Proposition 2

By definition of $\mathbb{K}(\hat{\pi})$ we have

$$V(\hat{d}_{zw}^{\hat{\pi}}) = V(\hat{d}_{zw}^{\bar{\pi}}) - \mathbb{K}(\hat{\pi}). \tag{14}$$

Then from

$$\mathbb{G}(\bar{\pi}) = \min\{V(\hat{d}_{zw}^{\bar{\pi}}) - V(\hat{d}_z), V(\hat{d}_{zw}^{\bar{\pi}}) - V(\hat{d}_w)\},$$

we can see

$$\max\{V(\hat{d}_z), V(\hat{d}_w)\} = V(\hat{d}_{zw}^{\bar{\pi}}) - \mathbb{G}(\bar{\pi}). \tag{15}$$

Finally, combining (14) and (15), we have

$$V(\hat{d}_{zw}^{\hat{\pi}}) = \max\{V(\hat{d}_z), V(\hat{d}_w)\} - \mathbb{K}(\hat{\pi}) + \mathbb{G}(\bar{\pi}).$$

As $V(\hat{d}_{z \cup w}) = \max\{V(\hat{d}_z), V(\hat{d}_w)\}$, we conclude that

$$V(\hat{d}_{zw}^{\hat{\pi}}) = \max\{V(\hat{d}_z), V(\hat{d}_w)\} - \mathbb{K}(\hat{\pi}) + \mathbb{G}(\bar{\pi}) = V(\hat{d}_{z \cup w}) - \mathbb{K}(\hat{\pi}) + \mathbb{G}(\bar{\pi}).$$

# G  Asymptotics of $\mathbb{K}(\hat{\pi})$

Throughout this section, we assume that $X \in [0,1]^p$ has a bounded density $f(x)$ and $\max\{|Y|, ||h||_\infty, ||q||_\infty\} \leq M$ for some $M > 0$. In addition, we assume that $\sup_{w,a,x} |\hat{h}(w,a,x) - h(w,a,x)| = o_p(n^{-\alpha})$, $\sup_{z,a,x} |\hat{q}(z,a,x) - q(z,a,x)| = o_p(n^{-\beta})$ for some $\alpha, \beta > 0$ (Chen and Christensen, 2013). Given the training dataset, we define an oracle estimator of $\delta(x; \hat{d}_z, \hat{d}_w)$

$$\delta'(x; \hat{d}_z, \hat{d}_w) \triangleq \frac{\sum_{i=1}^n \{h(W_i, \hat{d}_z(x, Z_i), x) - Y_i q(Z_i, A_i, x) \mathbb{I}\{\hat{d}_w(x, W_i) = A_i\}\} K(\frac{||x - X_i||}{\gamma})}{\sum_{i=1}^n K(\frac{||x - X_i||}{\gamma})}.$$

We assume that with probability larger than $1 - 1/n$, for any $d_z \in \mathcal{D}_{\mathcal{Z}}$ and $d_w \in \mathcal{D}_{\mathcal{W}}$, $\sup_x |\delta(x; d_z, d_w) - \delta'(x; d_z, d_w)| \leq C_1 n^{-\gamma}$ for some $C_1 > 0$ and $\gamma > 0$ under certain conditions (Jiang, 2017). If we further impose a restriction on the carnality of preliminary policy classes and assume $|\mathcal{D}_{\mathcal{Z}}| = o(n)$ and $|\mathcal{D}_{\mathcal{W}}| = o(n)$, by a straightforward calculation, we have $\sup_{x, d_z \in \mathcal{D}_{\mathcal{Z}}, d_w \in \mathcal{D}_{\mathcal{W}}} |\hat{\delta}(x; d_z, d_w) - \delta(x; d_z, d_w)| \leq C_2 n^{-\zeta}$ on a set $\mathcal{X}_0$ and $\mathbb{P}(\mathcal{X}_0^c) \to 0$, where $C_2 > 0, \zeta = \min\{\alpha, \beta, \gamma\}$, and $\mathcal{X}_0^c$ is the complement of $\mathcal{X}_0$.

To streamline the presentation, in the following, we abbreviate $\delta(X; \hat{d}_z, \hat{d}_w)$, $\delta'(X; \hat{d}_z, \hat{d}_w)$ and $\hat{\delta}(X; \hat{d}_z, \hat{d}_w)$ as $\delta(X), \delta'(X)$ and $\hat{\delta}(X)$, respectively. Two subsets of $\mathcal{X}$, namely $\mathcal{X}_{f1}$ and $\mathcal{X}_{f2}$, are defined as

$$\mathcal{X}_{f1} = \{x \in \mathcal{X} : \mathbb{I}\{\hat{\delta}(x) \geq 0\} = 1, \mathbb{I}\{\delta(x) \geq 0\} = 0\},$$
$$\mathcal{X}_{f2} = \{x \in \mathcal{X} : \mathbb{I}\{\hat{\delta}(x) \geq 0\} = 0, \mathbb{I}\{\delta(x) \geq 0\} = 1\},$$

and we also define the complement set $\mathcal{X}_c$ as

$$\mathcal{X}_c = \{x \in \mathcal{X} : \text{sign}(\hat{\delta}(x)) = \text{sign}(\delta(x))\},$$

with $\text{sign}(0) = 1$. We see that $\mathcal{X}_{f1} \cap \mathcal{X}_{f2} = \emptyset$, $\mathcal{X}_{f1} \cap \mathcal{X}_c = \emptyset$, $\mathcal{X}_{f2} \cap \mathcal{X}_c = \emptyset$, and $\mathcal{X}_{f1} \cup \mathcal{X}_{f2} \cup \mathcal{X}_c = \mathcal{X}$. From the definition of $\mathbb{K}(\hat{\pi})$, we have

$$\mathbb{K}(\hat{\pi}) = \int_{x \in \mathcal{X}} \mathbb{E}[\bar{\pi}(X; \hat{d}_z, \hat{d}_w) h(W, \hat{d}_z(X, Z), X) + (1 - \bar{\pi}(X; \hat{d}_z, \hat{d}_w)) Y q(Z, A, X) \mathbb{I}\{\hat{d}_w(X, W) = A\}$$

$$- \hat{\pi}(X; \hat{d}_z, \hat{d}_w) h(W, \hat{d}_z(X, Z), X) + (1 - \hat{\pi}(X; \hat{d}_z, \hat{d}_w)) Y q(Z, A, X) \mathbb{I}\{\hat{d}_w(X, W) = A\} | X = x] f(x) dx$$

$$= \int_{x \in \mathcal{X}_{f1}} -\delta(x) f(x) dx + \int_{x \in \mathcal{X}_{f2}} \delta(x) f(x) dx + \int_{x \in \mathcal{X}_c} 0 f(x) dx$$

$$= -\int_{x \in \mathcal{X}_{f1}} \delta(x) f(x) dx + \int_{x \in \mathcal{X}_{f2}} \delta(x) f(x) dx.$$

The second equation holds because if $x \in \mathcal{X}_{f1}$, $\hat{\pi}(x; \hat{d}_z, \hat{d}_w) = 1, \bar{\pi}(x; \hat{d}_z, \hat{d}_w) = 0$ and

$$\mathbb{E}[\bar{\pi}(X; \hat{d}_z, \hat{d}_w)h(W, \hat{d}_z(X, Z), X) + (1 - \bar{\pi}(X; \hat{d}_z, \hat{d}_w))Yq(Z, A, X)\mathbb{I}\{\hat{d}_w(X, W) = A\}$$

$$-\hat{\pi}(X; \hat{d}_z, \hat{d}_w)h(W, \hat{d}_z(X, Z), X) + (1 - \hat{\pi}(X; \hat{d}_z, \hat{d}_w))Yq(Z, A, X)\mathbb{I}\{\hat{d}_w(X, W) = A\}|X = x] = -\delta(x);$$

if $x \in \mathcal{X}_{f2}$, $\hat{\pi}(x; \hat{d}_z, \hat{d}_w) = 0, \bar{\pi}(x_f; \hat{d}_z, \hat{d}_w) = 1$ and

$$\mathbb{E}[\bar{\pi}(X; \hat{d}_z, \hat{d}_w)h(W, \hat{d}_z(X, Z), X) + (1 - \bar{\pi}(X; \hat{d}_z, \hat{d}_w))Yq(Z, A, X)\mathbb{I}\{\hat{d}_w(X, W) = A\}$$

$$-\hat{\pi}(X; \hat{d}_z, \hat{d}_w)h(W, \hat{d}_z(X, Z), X) + (1 - \hat{\pi}(X; \hat{d}_z, \hat{d}_w))Yq(Z, A, X)\mathbb{I}\{\hat{d}_w(X, W) = A\}|X = x] = \delta(x);$$

if $x \in \mathcal{X}_c$, $\hat{\pi}(x; \hat{d}_z, \hat{d}_w) = \bar{\pi}(x_f; \hat{d}_z, \hat{d}_w)$ and

$$\mathbb{E}[\bar{\pi}(X; \hat{d}_z, \hat{d}_w)h(W, \hat{d}_z(X, Z), X) + (1 - \bar{\pi}(X; \hat{d}_z, \hat{d}_w))Yq(Z, A, X)\mathbb{I}\{\hat{d}_w(X, W) = A\}$$

$$-\hat{\pi}(X; \hat{d}_z, \hat{d}_w)h(W, \hat{d}_z(X, Z), X) + (1 - \hat{\pi}(X; \hat{d}_z, \hat{d}_w))Yq(Z, A, X)\mathbb{I}\{\hat{d}_w(X, W) = A\}|X = x] = 0.$$

Therefore, we essentially need to bound $-\int_{x \in \mathcal{X}_{f1}} \delta(x)f(x)dx$ and $\int_{x \in \mathcal{X}_{f2}} \delta(x)f(x)dx$ follows a similar proof. In this regard, we further split $\mathcal{X}_{f1}$ to $\mathcal{X}_{f1,1} = \{x \in \mathcal{X}_{f1} : \delta(x) \in (-Cn^{-\varsigma}, Cn^{-\varsigma})\}$ and $\mathcal{X}_{f1,2} = \{x \in \mathcal{X}_{f1} : \delta(x) \notin (-Cn^{-\varsigma}, Cn^{-\varsigma})\}$. Then it is easy to see that $-\int_{x \in \mathcal{X}_{f1,1}} \delta(x)f(x)dx$ is bounded by $O(n^{-\varsigma})$ and $\mathbb{P}(\mathcal{X}_{f1,2})$ converges to 0 as $\mathbb{P}(\mathcal{X}_0^c)$ converges to 0. We then conclude that $\mathbb{K}(\hat{\pi}) = o(1)$ almost surely.

## H    Proof of Proposition 3

We start with defining two subsets of $\mathcal{X}$,

$$\mathcal{X}_{g1} = \{x \in \mathcal{X} : \bar{\pi}(x, \hat{d}_z, \hat{d}_w) = 1, \bar{\pi}(x, d_z^*, d_w^*) = 0\},$$
$$\mathcal{X}_{g2} = \{x \in \mathcal{X} : \bar{\pi}(x, \hat{d}_z, \hat{d}_w) = 0, \bar{\pi}(x, d_z^*, d_w^*) = 1\},$$

and we also define the complement set $\mathcal{X}_{gc}$ as

$$\mathcal{X}_{gc} = \{x \in \mathcal{X} : \bar{\pi}(x, \hat{d}_z, \hat{d}_w) = \bar{\pi}(x, d_z^*, d_w^*)\},$$

which can also be split into

$$\mathcal{X}_{gc1} = \{x \in \mathcal{X} : \bar{\pi}(x, \hat{d}_z, \hat{d}_w) = \bar{\pi}(x, d_z^*, d_w^*) = 0\},$$
$$\mathcal{X}_{gc2} = \{x \in \mathcal{X} : \bar{\pi}(x, \hat{d}_z, \hat{d}_w) = \bar{\pi}(x, d_z^*, d_w^*) = 1\}.$$

We see that $\mathcal{X}_{g1} \cap \mathcal{X}_{g2} = \emptyset, \mathcal{X}_{gc1} \cap \mathcal{X}_{gc2} = \emptyset, \mathcal{X}_{gc1} \cup \mathcal{X}_{gc2} = \mathcal{X}_{gc}, \mathcal{X}_{g1} \cap \mathcal{X}_{gc} = \emptyset, \mathcal{X}_{g2} \cap \mathcal{X}_{gc} = \emptyset$, and $\mathcal{X}_{g1} \cup \mathcal{X}_{g2} \cup \mathcal{X}_{gc} = \mathcal{X}$.

From the definition of $V(d_{zw}^{\bar{\pi}*})$ and $V(\hat{d}_{zw}^{\bar{\pi}})$, we have

$$V(d_{zw}^{\bar{\pi}*}) - V(\hat{d}_{zw}^{\bar{\pi}}) = \int_{x \in \mathcal{X}} \mathbb{E}[\bar{\pi}(X; d_z^*, d_w^*)h(W, d_z^*(X, Z), X) + (1 - \bar{\pi}(X; d_z^*, d_w^*))Yq(Z, A, X)\mathbb{I}\{d_w^*(X, W) = A\}$$

$$- \bar{\pi}(X; \hat{d}_z, \hat{d}_w)h(W, \hat{d}_z(X, Z), X) - (1 - \bar{\pi}(X; \hat{d}_z, \hat{d}_w))Yq(Z, A, X)\mathbb{I}\{\hat{d}_w(X, W) = A\}|X = x]f(x)dx$$

$$= \int_{x \in \mathcal{X}_{g1}} E[Yq(Z, A, X)\mathbb{I}\{d_w^*(X, W) = A\} - h(W, \hat{d}_z(X, Z), X)|X = x]f(x)dx$$

$$+ \int_{x \in \mathcal{X}_{g2}} E[h(W, d_z^*(X, Z), X) - Yq(Z, A, X)\mathbb{I}\{\hat{d}_w(X, W) = A\}|X = x]f(x)dx$$

$$+ \int_{x \in \mathcal{X}_{gc1}} E[Yq(Z, A, X)\mathbb{I}\{d_w^*(X, W) = A\} - Yq(Z, A, X)\mathbb{I}\{\hat{d}_w(X, W) = A\}|X = x]f(x)dx$$

$$+ \int_{x \in \mathcal{X}_{gc2}} E[h(W, d_z^*(X, Z), X) - h(W, \hat{d}_z(X, Z), X)|X = x]f(x)dx.$$

Then it is easy to see that

$$\int_{x \in \mathcal{X}_{gc1}} E[Yq(Z, A, X)\mathbb{I}\{d_w^*(X, W) = A\} - Yq(Z, A, X)\mathbb{I}\{\hat{d}_w(X, W) = A\}|X = x]f(x)dx$$

and

$$\int_{x \in \mathcal{X}_{gc2}} E[h(W, d_z^*(X, Z), X) - h(W, \hat{d}_z(X, Z), X)|X = x]f(x)dx$$

converge to 0 in probability according to Assumption 6.

Therefore, we essentially need to bound

$$\int_{x \in \mathcal{X}_{g1}} E[Yq(Z, A, X)\mathbb{I}\{d_w^*(X, W) = A\} - h(W, \hat{d}_z(X, Z), X)|X = x]f(x)dx$$

and

$$\int_{x \in \mathcal{X}_{g2}} E[h(W, d_z^*(X, Z), X) - Yq(Z, A, X)\mathbb{I}\{\hat{d}_w(X, W) = A\}|X = x]f(x)dx.$$

We further split $\mathcal{X}_{g1}$ to $\mathcal{X}_{g1,1} = \{x \in \mathcal{X}_{g1} : E[Yq(Z, A, X)\mathbb{I}\{d_w^*(X, W) = A\} - h(W, \hat{d}_z(X, Z), X)|X = x] \in (-Cn^{-\eta}, Cn^{-\eta})\}$ and $\mathcal{X}_{g1,2} = \{x \in \mathcal{X}_{g1} : E[Yq(Z, A, X)\mathbb{I}\{d_w^*(X, W) = A\} - h(W, \hat{d}_z(X, Z), X)|X = x] \notin (-Cn^{-\eta}, Cn^{-\eta})\}$ where $\eta = \min\{\xi, \varphi\}$. Then it is easy to see that

$$\int_{x \in \mathcal{X}_{g1,1}} E[Yq(Z, A, X)\mathbb{I}\{d_w^*(X, W) = A\} - h(W, \hat{d}_z(X, Z), X)|X = x]f(x)dx$$

is bounded by $O(n^{-\eta})$ and $\mathbb{P}(\mathcal{X}_{g1,2})$ converges to 0 in probability based on Assumption 6 and the definition of $\mathcal{X}_{g1}$. A similar proof can also be conducted to obtain $\int_{x \in \mathcal{X}_{g2}} E[h(W, d_z^*(X, Z), X) - Yq(Z, A, X)\mathbb{I}\{\hat{d}_w(X, W) = A\}|X = x]f(x)dx$ is small enough. We then have that $V(\hat{d}_{zw}^{\bar{\pi}}) \xrightarrow{p} V(d_{zw}^{\bar{\pi}*})$.

As we have proved that $\mathbb{K}(\hat{\pi}) = V(\hat{d}_{zw}^{\bar{\pi}}) - V(\hat{d}_{zw}^{\hat{\pi}}) = o(1)$ almost surely in Appendix G, we finally conclude that $V(\hat{d}_{zw}^{\hat{\pi}}) \xrightarrow{p} V(d_{zw}^{\bar{\pi}*})$.

## I   Data generating mechanisim and parameter setup in Section 4

The data generating mechanism for $(X, A, Z, W, U)$ is summarized in Table 1, and the setups of varying parameters in each scenario are summarized in Table 2.

## J   Derivation of optimal ITRs considered in Section 4

From

$$(Z, W, U)|A, X \sim \mathcal{N}\left\{ \left( \begin{array}{c} \alpha_0 + \alpha_a \frac{1+A}{2} + \alpha_x X \\ \mu_0 + \mu_a \frac{1+A}{2} + \mu_x X \\ \kappa_0 + \kappa_a \frac{1+A}{2} + \kappa_x X \end{array} \right), \Sigma = \left( \begin{array}{ccc} \sigma_z^2 & \sigma_{zw} & \sigma_{zu} \\ \sigma_{zw} & \sigma_w^2 & \sigma_{wu} \\ \sigma_{zu} & \sigma_{wu} & \sigma_u^2 \end{array} \right) \right\},$$

and

$$(Z, W, U)|X \sim \mathcal{N}\left\{ \left( \begin{array}{c} \alpha_0 + \alpha_a \mathbb{P}(A = 1|X) + \alpha_x X \\ \mu_0 + \mu_a \mathbb{P}(A = 1|X) + \mu_x X \\ \kappa_0 + \kappa_a \mathbb{P}(A = 1|X) + \kappa_x X \end{array} \right), \Sigma = \left( \begin{array}{ccc} \sigma_z^2 & \sigma_{zw} & \sigma_{zu} \\ \sigma_{zw} & \sigma_w^2 & \sigma_{wu} \\ \sigma_{zu} & \sigma_{wu} & \sigma_u^2 \end{array} \right) \right\},$$

the following results hold,

$$\mathbb{E}[W|X, A, U] = \mu_0 + \mu_a \frac{1+A}{2} + \mu_x X + \frac{\sigma_{wu}}{\sigma_u^2}(U - \kappa_0 - \kappa_a \frac{1+A}{2} - \kappa_x X), \tag{16}$$

$$\mathbb{E}[U|X, Z] = \kappa_0 + \kappa_a \mathbb{P}(A = 1|X) + \kappa_x X + \frac{\sigma_{zu}}{\sigma_z^2}(Z - \alpha_0 - \alpha_a \mathbb{P}(A = 1|X) - \alpha_x X), \tag{17}$$

$$\mathbb{E}[U|X, W] = \kappa_0 + \kappa_a \mathbb{P}(A = 1|X) + \kappa_x X + \frac{\sigma_{wu}}{\sigma_w^2}(W - \mu_0 - \mu_a \mathbb{P}(A = 1|X) - \mu_x X). \tag{18}$$

## Generating Mechanism — Fixed Parameter Setting

| Variables | Generating Mechanism | Fixed Parameter Setting |
|---|---|---|
| $X \in \mathbb{R}^2$ | $X \sim \mathcal{N}(\Gamma_x, \Sigma_x)$ | $\Gamma_x = (0.25, 0.25)^T,\ \Sigma_x = \begin{pmatrix} 0.25^2 & 0 \\ 0 & 0.25^2 \end{pmatrix}$ |
| $A \in \{1, -1\}$ | $\left(\frac{A+1}{2}\right) \mid X \sim \mathrm{Bern}(p_a)$ | $p_a = \frac{1}{1+\exp\{(0.125, 0.125)^T X\}}$ |
| $Z \in \mathbb{R}$ $W \in \mathbb{R}$ $U \in \mathbb{R}$ | $(Z, W, U)\mid A, X \sim \mathcal{N}\left\{ \begin{pmatrix} \alpha_0 + \alpha_a \frac{1+A}{2} + \alpha_x X \\ \mu_0 + \mu_a \frac{1+A}{2} + \mu_x X \\ \kappa_0 + \kappa_a \frac{1+A}{2} + \kappa_x X \end{pmatrix},\ \Sigma = \begin{pmatrix} \sigma_z^2 & \sigma_{zw} & \sigma_{zu} \\ \sigma_{zw} & \sigma_w^2 & \sigma_{wu} \\ \sigma_{zu} & \sigma_{wu} & \sigma_u^2 \end{pmatrix} \right\}$ | $\alpha_0 = \alpha_a = \mu_0 = \kappa_0 = \kappa_a = \sigma_{zw} = 0.25,$ $\mu_a = 0.125,\ \alpha_x = \mu_x = \kappa_x = (0.25, 0.25)^T,$ $\sigma_{zu} = \sigma_{wu} = 0.5,\ \sigma_z = \sigma_w = \sigma_u = 1$ |
| $Y \in \mathbb{R}$ | $Y \sim \mathcal{N}\{\mathbb{E}(Y\mid W, U, A, Z, X), \sigma_y^2\}$ | $\sigma_y = 0.25,\ b_0 = 2,\ \omega = 2$ |

* As for generation of $Y$, $\mathbb{E}(Y\mid X, A, Z, W, U) = b_0 + b_1(X)\frac{1+A}{2} + b_2(X)X + \left(b_w + b_a\frac{1+A}{2} + b_3(X)A + (b_w + b_a\frac{1+A}{2} + b_3(X)A - \omega\right)\mathbb{E}(W\mid U, X) + \omega W$, where $\mathbb{E}(W\mid U, X) = \mu_0 + \mu_x X + \frac{\sigma_{wu}}{\sigma_u^2}(U - \kappa_0 - \kappa_x X)$.

Table 1: Data generating mechanism and setup for fixed parameters across scenarios.

| Scenario | Parameter Setup | | | | |
|---|---|---|---|---|---|
| **Number** | $b_1(X)$ | $b_2(X)$ | $b_3(X)$ | $b_a$ | $b_w$ |
| **1** | $0.5 + 3X_{(1)} - 5X_{(2)}$ | $(0.25, 0.25)^T$ | $0$ | $0.25$ | $8$ |
| **2** | $0.5 + 3X_{(1)} - 5X_{(2)}$ | $(0.25, 0.25)^T$ | $0$ | $0$ | $8$ |
| **3** | $2.3 + \|X_{(1)} - 1\| - \|X_{(2)} + 1\|$ | $X^T$ | $\sin(X_{(1)}) - 2\cos(X_{(2)})$ | $-2.5$ | $4$ |
| **4** | $0.25 - 6X_{(1)}X_{(2)}$ | $X^T$ | $0$ | $0$ | $5$ |
| **5** | $0.1 - 2X_{(1)}^2$ | $X^T$ | $4X_{(2)}^2$ | $0.8$ | $8$ |
| **6** | $-0.5 + \exp(X_{(1)}) - 3X_{(2)}$ | $(0.25, 0.25)^T$ | $0$ | $0$ | $8$ |

*$X_{(1)}, X_{(2)}$ denote the first and second dimensions of $X$.
* The parameter settings in scenarios 1-4 are considered by Qi et al. (2023).

Table 2: The varying parameters for each scenario.

Recall that

$$\mathbb{E}(Y|X, A, Z, W, U) = b_0 + b_1(X)\frac{1+A}{2} + b_2(X)X + \left(b_w + b_a\frac{1+A}{2} + b_3(X)A - \omega\right)$$
$$\left(\mu_0 + \mu_x X + \frac{\sigma_{wu}}{\sigma_u^2}(U - \kappa_0 - \kappa_x X)\right) + \omega W,$$

then we can find that

$$\mathbb{E}(Y|X, A, Z, U) = b_0 + b_1(X)\frac{1+A}{2} + b_2(X)X + \left(b_w + b_a\frac{1+A}{2} + b_3(X)A - \omega\right)$$
$$\left(\mu_0 + \mu_x X + \frac{\sigma_{wu}}{\sigma_u^2}(U - \kappa_0 - \kappa_x X)\right) + \omega\mathbb{E}[W|X, A, Z, U],$$
$$= b_0 + b_1(X)\frac{1+A}{2} + b_2(X)X + \left(b_w + b_a\frac{1+A}{2} + b_3(X)A - \omega\right)$$
$$\left(\mu_0 + \mu_x X + \frac{\sigma_{wu}}{\sigma_u^2}(U - \kappa_0 - \kappa_x X)\right) + \omega\mathbb{E}[W|X, A, U],$$
$$= b_0 + b_1(X)\frac{1+A}{2} + b_2(X)X + \left(b_w + b_a\frac{1+A}{2} + b_3(X)A - \omega\right)$$
$$\left(\mu_0 + \mu_x X + \frac{\sigma_{wu}}{\sigma_u^2}(U - \kappa_0 - \kappa_x X)\right) + \omega\left(\mu_0 + \mu_x X + \frac{\sigma_{wu}}{\sigma_u^2}(U - \kappa_0 - \kappa_x X)\right),$$
$$= b_0 + b_1(X)\frac{1+A}{2} + b_2(X)X + \left(b_w + b_a\frac{1+A}{2} + b_3(X)A\right)\left(\mu_0 + \mu_x X + \frac{\sigma_{wu}}{\sigma_u^2}(U - \kappa_0 - \kappa_x X)\right),$$
$$(19)$$

where the first equality is duo to Assumption 1, and the second equality is due to (16), and

$$\mathbb{E}(Y|X, A, W, U) = \mathbb{E}(Y|X, A, Z, W, U)$$
$$= b_0 + b_1(X)\frac{1+A}{2} + b_2(X)X + \left(b_w + b_a\frac{1+A}{2} + b_3(X)A - \omega\right)$$
$$\left(\mu_0 + \mu_x X + \frac{\sigma_{wu}}{\sigma_u^2}(U - \kappa_0 - \kappa_x X)\right) + \omega W, \qquad (20)$$

where the first equality is due to Assumption 1. Furthermore, note that

$$\mathbb{E}[h(W, 1, X)|X, Z, U] = \mathbb{E}[h(W, 1, X)|X, U]$$
$$= \mathbb{E}[Y|X, A = 1, U]$$
$$= \mathbb{E}[Y|X, A = 1, Z, U]$$
$$= b_0 + b_1(X) + b_2(X)X + (b_w + b_a + b_3(X))\left(\mu_0 + \mu_x X + \frac{\sigma_{wu}}{\sigma_u^2}(U - \kappa_0 - \kappa_x X)\right),$$

where the first and third equality is due to Assumption 1, the second equality follows from Theorem 1 of Miao et al. (2018a) under Assumptions 2 and 3, and the last equality is by (19). Similarly,

$$\mathbb{E}[h(W, -1, X)|X, Z, U] = \mathbb{E}[Y|X, A = -1, Z, U]$$
$$= b_0 + b_2(X)X + (b_w - b_3(X))\left(\mu_0 + \mu_x X + \frac{\sigma_{wu}}{\sigma_u^2}(U - \kappa_0 - \kappa_x X)\right).$$

On the other hand,

$$\mathbb{E}[Yq(Z, 1, X)\mathbb{I}\{A = 1\}|X, W, U] = \mathbb{P}(A = 1|X, W, U)\mathbb{E}[Yq(Z, 1, X)|X, A = 1, W, U]$$
$$= \mathbb{P}(A = 1|X, U)\mathbb{E}[q(Z, 1, X)|X, A = 1, U]\mathbb{E}[Y|X, A = 1, W, U]$$
$$= \mathbb{E}[Y|X, A = 1, W, U]$$
$$= b_0 + b_1(X) + b_2(X)X + (b_w + b_a + b_3(X) - \omega)$$
$$\left(\mu_0 + \mu_x X + \frac{\sigma_{wu}}{\sigma_u^2}(U - \kappa_0 - \kappa_x X)\right) + \omega W,$$

where the second equality is due to Assumption 1, and the third equality is due to Theorem 2.2 of Cui et al. (2023) under Assumptions 4 and 5, and the last equality is due to (20). Similarly,

$$\mathbb{E}[Yq(Z, -1, X)\mathbb{I}\{A = -1\}|X, W, U] = \mathbb{E}[Y|X, A = -1, W, U]$$
$$= b_0 + b_2(X)X + (b_w - b_3(X) - \omega)$$
$$\left(\mu_0 + \mu_x X + \frac{\sigma_{wu}}{\sigma_u^2}(U - \kappa_0 - \kappa_x X)\right) + \omega W.$$

Then we can find that

$$\mathbb{E}[h(W, 1, X) - h(W, -1, X)|X, Z, U] = b_1(X) + (b_a + 2b_3(X))\left(\mu_0 + \mu_x X + \frac{\sigma_{wu}}{\sigma_u^2}(U - \kappa_0 - \kappa_x X)\right),$$

$$\mathbb{E}[Yq(Z, 1, X)\mathbb{I}\{A = 1\} - Yq(Z, -1, X)\mathbb{I}\{A = -1\}|X, W, U]$$
$$= b_1(X) + (b_a + 2b_3(X))\left(\mu_0 + \mu_x X + \frac{\sigma_{wu}}{\sigma_u^2}(U - \kappa_0 - \kappa_x X)\right).$$

Furthermore, we have

$$\mathbb{E}[h(W, 1, X) - h(W, -1, X)|X, Z] = \mathbb{E}[\mathbb{E}[h(W, 1, X) - h(W, -1, X)|X, Z, U]]$$
$$= b_1(X) + (b_a + 2b_3(X))\left(\mu_0 + \mu_x X + \frac{\sigma_{wu}}{\sigma_u^2}(\mathbb{E}[U|X, Z] - \kappa_0 - \kappa_x X)\right), \tag{21}$$

$$\mathbb{E}[Yq(Z, 1, X)\mathbb{I}\{A = 1\} - Yq(Z, -1, X)\mathbb{I}\{A = -1\}|X, W]$$
$$= \mathbb{E}[\mathbb{E}[Yq(Z, 1, X)\mathbb{I}\{A = 1\} - Yq(Z, -1, X)\mathbb{I}\{A = -1\}|X, W, U]]$$
$$= b_1(X) + (b_a + 2b_3(X))\left(\mu_0 + \mu_x X + \frac{\sigma_{wu}}{\sigma_u^2}(\mathbb{E}[U|X, W] - \kappa_0 - \kappa_x X)\right). \tag{22}$$

Therefore, plug (17) and (18) into (21) and (22) respectively, we can find that

$$\mathbb{E}[h(W, 1, X) - h(W, -1, X)|X, Z] = b_1(X) + (b_a + 2b_3(X))(\mu_0 + \mu_x X + \frac{\sigma_{wu}}{\sigma_u^2}(\kappa_0 + \kappa_a \mathbb{P}(A = 1|X)$$
$$+ \kappa_x X + \frac{\sigma_{zu}}{\sigma_z^2}(Z - \alpha_0 - \alpha_a \mathbb{P}(A = 1|X) - \alpha_x X) - \kappa_0 - \kappa_x X)),$$

$$\mathbb{E}[Yq(Z, 1, X)\mathbb{I}\{A = 1\} - Yq(Z, -1, X)\mathbb{I}\{A = -1\}|X, W]$$
$$= b_1(X) + (b_a + 2b_3(X))(\mu_0 + \mu_x X + \frac{\sigma_{wu}}{\sigma_u^2}(\kappa_0 + \kappa_a \mathbb{P}(A = 1|X)$$
$$+ \kappa_x X + \frac{\sigma_{wu}}{\sigma_w^2}(W - \mu_0 - \mu_a \mathbb{P}(A = 1|X) - \mu_x X) - \kappa_0 - \kappa_x X)).$$

Hence,

$$
\begin{aligned}
d_z^*(X, Z) &= \text{sign}\{\mathbb{E}[h(W, 1, X) - h(W, -1, X)|X, Z]\} \\
&= \text{sign}\{b_1(X) + (b_a + 2b_3(X))(\mu_0 + \mu_x X + \frac{\sigma_{wu}}{\sigma_u^2}(\kappa_0 + \kappa_a \mathbb{P}(A = 1|X) \\
&\quad + \kappa_x X + \frac{\sigma_{zu}}{\sigma_z^2}(Z - \alpha_0 - \alpha_a \mathbb{P}(A = 1|X) - \alpha_x X) - \kappa_0 - \kappa_x X))\}, \\
d_w^*(X, W) &= \text{sign}\{\mathbb{E}[Yq(Z, 1, X)\mathbb{I}\{A = 1\} - Yq(Z, -1, X)\mathbb{I}\{A = -1\}|X, W]\} \\
&= \text{sign}\{b_1(X) + (b_a + 2b_3(X))(\mu_0 + \mu_x X + \frac{\sigma_{wu}}{\sigma_u^2}(\kappa_0 + \kappa_a \mathbb{P}(A = 1|X) \\
&\quad + \kappa_x X + \frac{\sigma_{wu}}{\sigma_w^2}(W - \mu_0 - \mu_a \mathbb{P}(A = 1|X) - \mu_x X) - \kappa_0 - \kappa_x X))\}.
\end{aligned}
$$

## K   Implementation details of numerical experiments

**Step (i)** The method we adopt is neural maximum moment restriction (NMMR), which employs multilayer perceptron (MLP) to estimate the confounding bridges (Kompa et al., 2022). The target loss functions are set as

$$
R(h) = \mathbb{E}[(Y - h(W, A, X))(Y' - h(W', A', X'))K_z((Z, A, X), (Z', A', X'))],
$$

$$
R(q, a) = \mathbb{E}[(1 - \mathbb{I}\{A = a\}q(Z, a, X))(1 - \mathbb{I}\{A' = a\}q(Z', a, X'))K_w((W, X), (W', X'))], \text{ for } a \in \mathcal{A},
$$

where $(Z', W', A', X', Y')$ are independent copies of $(Z, W, A, X, Y)$, and $K_z : (\mathcal{Z} \times \mathcal{A} \times \mathcal{X})^2 \to \mathbb{R}, K_w : (\mathcal{W} \times \mathcal{X})^2 \to \mathbb{R}$ denote continuous, bounded, and integrally strictly positive definite (ISPD) kernels. In practice, we use the empirical risk instead, i.e.,

$$
\hat{R}(h) = \frac{1}{n(n-1)} \sum_{i,j=1,i \neq j}^{n} (y_i - h_i)(y_j - h_j)k_{z,ij}, \tag{23}
$$

$$
\hat{R}(q, a) = \frac{1}{n(n-1)} \sum_{i,j=1,i \neq j}^{n} (1 - \mathbb{I}\{a_i = a\}q_i)(1 - \mathbb{I}\{a_j = a\}q_j)k_{w,ij}, \text{ for } a \in \mathcal{A}, \tag{24}
$$

where $h_i = h(w_i, a_i, x_i), q_i = (z_i, a_i, x_i), k_{z,ij} = K_z((z_i, a_i, x_i)(z_j, a_j, x_j))$ and $k_{w,ij} = K_w((w_i, x_i), (w_j, x_j))$. In addition, we add a penalty term with respect to network weights to avoid overfitting.

As for the hyperparameters tuning procedure, we consider employing multilayer perceptrons with 2-8 fully connected layers with a variable number of hidden units. We then perform a grid search over the following parameters: learning rate, penalty coefficient, number of epochs, batch size, depth of the network, and width of the network. For every permutation of these parameters, we train a network based on the determined architecture and parameter values. Subsequently, we compute the empirical risk. Our aim is to pinpoint the parameter combination that yields the lowest empirical risk. These identified optimal parameters are then utilized to construct a refined neural network, which, in turn, serves as the foundation for conducting estimations. The parameter setup is summarized in Table 3. For detailed insights into the specific hyperparameter choices and architectural dimensions, we refer to supplementary Section B in Kompa et al. (2022).

| Parameter | Value |
|---|---|
| Number of epoch | 150 |
| Batch size | 250 |
| Learning rate | 0.003 |
| Penalty coefficient | 0.001, 0.01, 0.1 |
| Depth of network | 4 (for estimating $h$) |
| | 8 (for estimating $q$) |
| Width of network | 80 |

Table 3: Parameter setup for step (i)

**Step (ii)** For the estimation of preliminary ITRs, we follow the main text to solve the proposed optimization problems. For instance, to estimate $d_z^*$, we solve the following optimization problem:

$$\hat{g}_z \in \arg\min_{g_z \in \mathcal{G}_z} \mathbb{P}_n[\{\hat{h}(W, 1, X) - \hat{h}(W, -1, X)\}\phi(g_z(X, Z))] + \rho_z\|g_z\|_{\mathcal{G}_z}^2.$$

Here, $g_z$ represents a measurable decision function in $\mathcal{G}_{\mathcal{Z}} : \mathcal{X} \times \mathcal{Z} \to \mathbb{R}$ used to indicate $d_z$ (e.g., $d_z(X, Z) = \text{sign}(g_z(X, Z)))$, $\phi$ denotes the hinge loss function $\phi(x) = \max\{1 - x, 0\}$, and $\rho_z > 0$ is a tuning parameter. As for the tuning procedure regarding $\rho_z$, when $g_z$ is treated as a linear rule, for each predefined $\rho_z$, the data is divided into $K$ folds. For each $k \in [K]$, we compute $\hat{h}^{(-k)}$ and $\hat{g}_z^{(-k)}$, and then calculate the empirical value using the validation data. By averaging the empirical values across $K$ folds for each value of $\rho_z$, we identify the parameter that maximizes the average empirical value. The finalized parameter is then employed to determine $\hat{g}_z$. Such a procedure can be extended. For example, when considering $g_z$ as a RKHS, it is advisable to apply the cross-fitting procedure separately for each combination of pre-defined $\rho_z$ and bandwidth, with details presented in Qi et al. (2023). And the estimation of $\hat{d}_w$ can be approached in a similar manner.

For more estimators regarding $d_z^*$ and $d_w^*$, we refer to Bennett and Kallus (2023); Sverdrup and Cui (2023); Wang et al. (2022). One could further expand the estimation pipeline utilized in unconfounded scenarios and leverage state-of-the-art machine learning techniques (Chen et al., 2020; Raghu et al., 2017; Yoon et al., 2018) to tackle the weighted classification problems and construct estimates.

**Step (iii)** The estimation of $\bar{\pi}$ follows the procedure given in the main text. As for the selection of bandwidth in the Nadaraya-Watson kernel regression estimator, we employ Scott's rule of thumb (Scott, 2015) and set $\gamma = 1.06\hat{\sigma}n^{-1/5}$, where $\hat{\sigma}$ is the estimated standard deviation of $X$. For more methods regarding estimation of $\delta(\cdot)$, we refer to Chen (2017); Dalmasso et al. (2020); Dinh et al. (2016); Sohn et al. (2015).

For the convenience of readers to reproduce the results, the pseudo-code of the whole pipeline is presented in Algorithm 1. The code of implementation can also be accessed on GitHub [2].

---

**Algorithm 1:** Estimation of optimal ITR $d_{zw}^{\bar{\pi}*}$

---

1  **Input**: Training data
2  Construct MLP models to estimate $h(w, a, x)$ and $q(z, a, x)$:
3  **Repeat** for different penalty coefficients:
4     **for** *each epoch* **do**
5        **for** *each batch* **do**
6           Compute loss function (23) and (24) based on the batch
7           Update the internal model parameter
8     **end**
9  **end**
10  **Finalize** the penalty coefficient which minimizes the empirical loss, and obtain $\hat{h}(w, a, x)$ and $\hat{q}(z, a, x)$
11  **Repeat** for different $\rho_z$ and $\rho_w$:
12     **for** *each batch* **do**
13        Find $\hat{g}_{z,b}, \hat{g}_{w,b}$ by (7) and (8) based on the $b$-th batch, estimated bridge functions, and specified $\rho_z$ and $\rho_w$, and then obtain $\hat{d}_{z,b}, \hat{d}_{w,b}$ based on $\hat{g}_{z,b}, \hat{g}_{w,b}$
14        Compute empirical value of $\hat{d}_{z,b}, \hat{d}_{w,b}$ respectively using the data not covered in the batch
15  **end**
16  **Finalize** $\rho_z$ and $\rho_w$ based on empirical values and then obtain $\hat{d}_z, \hat{d}_w$
17  Select bandwidth by Scott's rule of thumb
18  Find $\hat{\delta}(X; \hat{d}_z, \hat{d}_w)$ and then obtain $\hat{\pi}(X; \hat{d}_z, \hat{d}_w)$
19  **Output**: $\hat{d}_{zw}^{\hat{\pi}}$ constructed by (9)

---

---

[2] https://github.com/taoshen2022/Optimal-Treatment-Regimes-for-Proximal-Causal-Learning

# L  Additional results of numerical experiments

The experimental results with sample size $n = 500$ are presented in Figure 4. The experimental results with sample size $n = 500$ and an altered behavior policy (treatment is randomly assigned in this case) are presented in Figure 5.

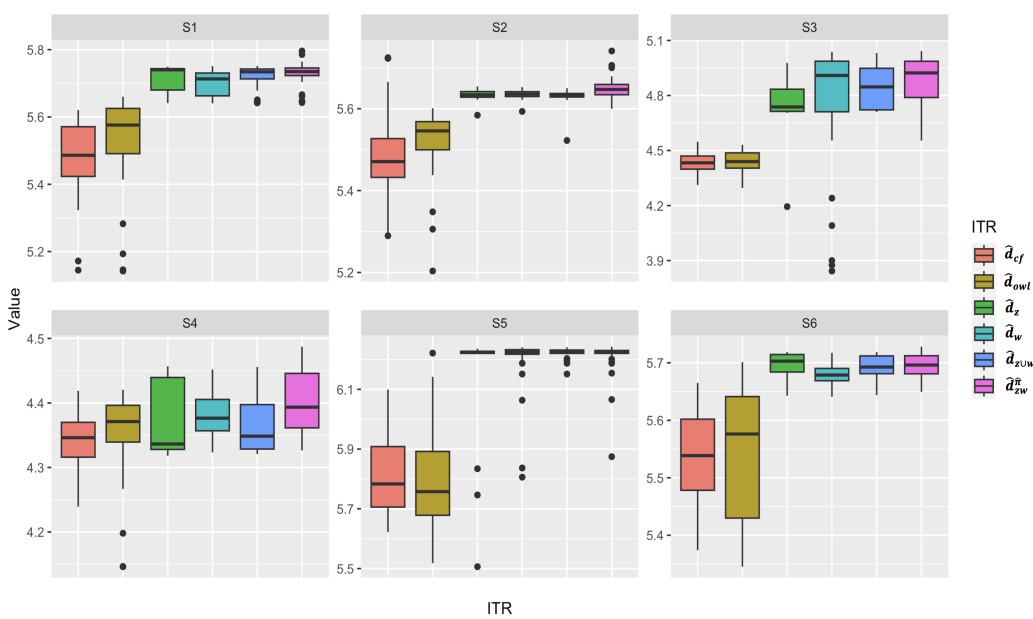

Figure 4: Boxplots of the empirical value functions with $n = 500$.

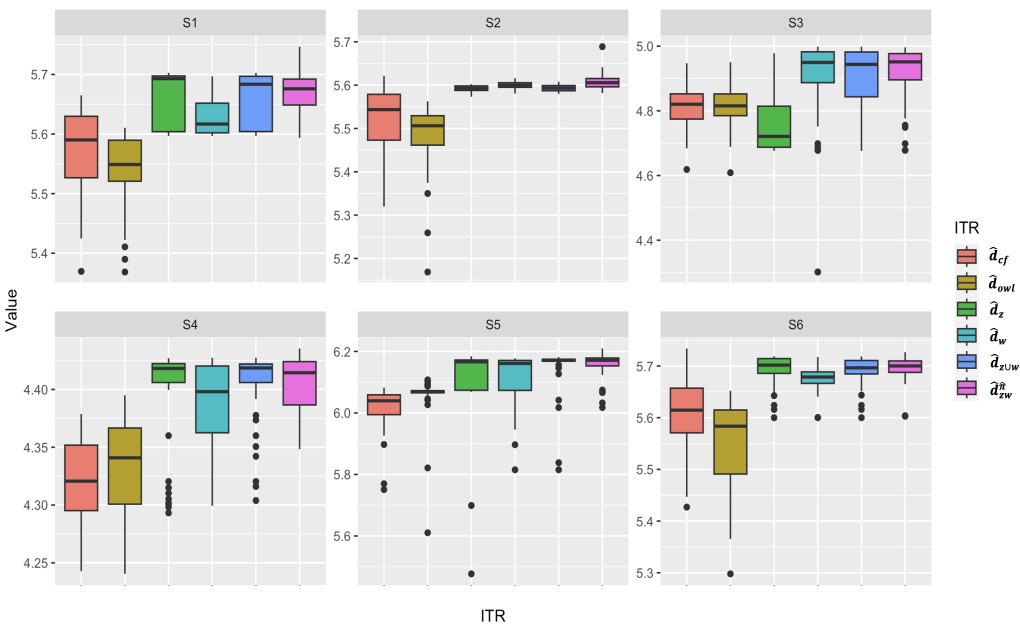

Figure 5: Boxplots of the empirical value functions with $n = 500$ and an altered behavior policy.

## M  Additional results of real data application

Regarding the quantitative analysis, Table 4 describes the estimated value functions of our proposed ITR, alongside existing approaches, under four settings with increasing numbers of proxies. For Setting 1, $Z = (pafi1, paco21), W = (ph1, hema1)$. For Setting 2, $Z = (pafi1, paco21, pot1), W = (ph1, hema1, bili1)$. For Setting 3, $Z = (pafi1, paco21, pot1, wt0), W = (ph1, hema1, bili1, sod1)$. For Setting 4, $Z = (pafi1, paco21, pot1, wt0, crea1), W = (ph1, hema1, bili1, sod1, alb1)$.

|  | $\hat{V}(\hat{d}_{cf})$ | $\hat{V}(\hat{d}_{owl})$ | $\hat{V}(\hat{d}_z)$ | $\hat{V}(\hat{d}_w)$ | $\hat{V}(\hat{d}_{z \cup w})$ | $\hat{V}(\hat{d}_{zw}^{\hat{\pi}})$ |
|---|---|---|---|---|---|---|
| Setting 1 | 24.84 | 24.97 | 25.12 | 26.61 | 27.86 | 28.21 |
|  | (3.06) | (2.93) | (4.69) | (3.34) | (2.28) | (3.28) |
| Setting 2 | 24.81 | 24.97 | 25.60 | 25.74 | 26.32 | 27.02 |
|  | (3.11) | (2.94) | (3.73) | (3.57) | (2.29) | (2.95) |
| Setting 3 | 24.79 | 24.97 | 26.12 | 25.53 | 26.76 | 27.83 |
|  | (3.02) | (2.93) | (3.61) | (3.29) | (2.76) | (3.03) |
| Setting 4 | 24.90 | 24.97 | 25.26 | 25.81 | 27.38 | 27.96 |
|  | (3.18) | (2.93) | (4.76) | (3.03) | (2.74) | (3.07) |

Table 4: Estimated values for different ITRs under different proxy variable settings.

As for the qualitative analysis, we present an illustrative example below. Regarding the estimated ITRs in Setting 1, the coefficient of $cat1\_lung$ is negative with a minor magnitude for $\hat{d}_z$, contrasting with a positive and relatively large coefficient observed for $\hat{d}_w$, which mirror the outcomes outlined in Qi et al. (2023). This finding suggests that, within the primary disease category of patients with lung cancer, $\hat{d}_z$ advocates for undergoing RHC, while $\hat{d}_w$ displays a notably inconclusive trend. As evidenced by $\hat{\pi}$, the prevailing trajectory for patients with $cat1\_lung = 1$ involves a strong inclination toward undergoing RHC, i.e., $\hat{\pi}(X) = 1$, aligning with the guidance offered by $\hat{d}_z$. Significantly, the domain knowledge underscores the potential for patients with advanced lung cancer to develop complications like pulmonary hypertension and coma, potentially warranting RHC for assessing pulmonary vascular changes and informing treatment strategies (Galie et al., 2009), which lends support to the recommendations offered by our proposed regime. Furthermore, it is important to note that the whole group of patients can be regarded as unions of multiple subgroups based on various distinct features, and the superiority of $\hat{d}_w$ is evident in some subgroups (e.g., $amihx$). These results show that our proposed ITR offers superior efficacy compared to $\hat{d}_z, \hat{d}_w$ and $\hat{d}_{z \cup w}$ as our methodology incorporates selection through $\hat{\pi}$.