# OpenReview forum: "Optimal Treatment Regimes for Proximal Causal Learning"
_NeurIPS.cc/2023/Conference — NeurIPS 2023 poster_

### Official Review · Reviewer_of5M · 2023-07-02

**Soundness:** 4 excellent
**Presentation:** 4 excellent
**Contribution:** 3 good
**Rating:** 5
**Confidence:** 5

**Summary:**

The authors present a new optimal individual treatment regime (ITR) within the proximal causal inference framework, which avoids the strong assumption of no unmeasured confounding. Instead, one assumes the effect of the unmeasured confounders flows exclusively through proxy variables, as defined through outcome-inducing and treatment-inducing confounding bridges. Compared to prior work, this optimal ITR that is defined with respect to a more flexible function class that depends on known confounders X, treatment-inducing confounding proxies Z, and outcome-inducing confounding proxies W.

**Strengths:**

The proposed ITR is a natural extension of existing ITRs by using a function \pi(x) that selectively chooses between two existing ITRs based on known confounders x. Under the proximal causal inference frameowrk, the proposed ITR is proven to be superior to existing ITRs in the literature. (Existing ITRs from Qi 2023 can be viewed as special cases of the proposed ITR.) The authors introduce a simple plugin estimator for the proposed ITR and show that the value of the resulting estimator is determined by approximation error of \pi and the gain from using \pi. Simulation studies show that the proposed ITR is either superior or comparable to existing ITRs. The manuscript is clearly written. The authors provide a nice review of prior work in this area and clearly describe how their work builds on existing work.

**Weaknesses:**

1. The proposed extension of the ITR function class appears quite incremental. The value of the proposed ITR follows directly from application of the tower rule. The paper would be greatly strengthened if the authors can show that this is the best one can do, e.g. showing that the value of a more complex ITR function class would be unidentifiable without much stronger assumptions.

2. In the simulation studies, the improvement in mean value when using the proposed optimal ITR over existing ITRs is large only in scenario 2. In all other scenarios, the improvement is small. Can the authors explain the behavior in this simulation study? Also, can the authors explain settings in which the proposed ITR is expected to substantially improve over existing ITRs? My guess is that the gain is biggest when (i) there are large differences between expected value at each X for the ITR with domain (X,W) and the ITR with domain (X,Z) and (ii) the optimal pi function has high variance (e.g. pi(X) is equal to 1 half of the time). Does this correspond to scenario 2?

3. The authors perform a real-data analysis in Section 5, which illustrates how the proposed ITR is different from existing ITRs. However, the authors do not calculate the values of the estimated ITR, so readers cannot compare the performance of the proposed ITR against existing ITRs. Do the authors have estimates of the values of the estimated ITRs?

4. The number of treatment-inducing confounding proxies and outcome-inducing confounding proxies were small in both the simulation studies and real-world data analysis. However, the practical appeal of the proximal causal inference framework is its use of proxies for unmeasured confounders, which would suggest the use of many variables as potential proxies. Can the authors include simulations that reflect more realistic settings where more proxy variables are used? How does the proposed method perform as the number of these proxies increases?

**Questions:**

1. How does one do statistical inference for the value of the estimated ITR?

**Limitations:**

The authors have not discussed limitations of the work.

---

> ### Author Rebuttal · Authors · 2023-08-09
>
> We sincerely appreciate the constructive suggestions from the reviewer.
>
> W1: The proposed extension of the ITR function class appears quite incremental. The value of the proposed ITR follows directly from application of the tower rule. The paper would be greatly strengthened if the authors can show that this is the best one can do, e.g. showing that the value of a more complex ITR function class would be unidentifiable without much stronger assumptions.
>
> R: Thank you. We have pointed out that $E[Y(a)|X,W,Z]$ is not identifiable under the current assumptions and framework (Qi et al. 2023 JASA), and our proposed class is much more general than the classes considered in previous literature. However, at this juncture, whether a more general policy class could be considered remains an open question. We add this important point in the discussion section.
>
> W2: In the simulation studies, the improvement in mean value when using the proposed optimal ITR over existing ITRs is large only in scenario 2. In all other scenarios, the improvement is small. Can the authors explain the behavior in this simulation study? Also, can the authors explain settings in which the proposed ITR is expected to substantially improve over existing ITRs? My guess is that the gain is biggest when (i) there are large differences between expected value at each X for the ITR with domain (X,W) and the ITR with domain (X,Z) and (ii) the optimal pi function has high variance (e.g. pi(X) is equal to 1 half of the time). Does this correspond to scenario 2?
>
>
> R: We agree with your insightful guessing. As we indicate in Proposition 1, the value of the proposed estimator has a significant improvement compared to other optimal ITRs depending on the magnitude of $G(\bar{\pi})$, which is related to disparities between expected values at each $X$ and the variance of $\bar{\pi}$. A pertinent illustration can be drawn from our simulation results; for instance, in scenarios 1, 2, 4 & 6, the variance of $\hat{\pi}$ is large. On the other hand, scenarios 3 & 5 portray relatively limited variation in the two estimated expected values for each $X$.
>
> W3: The authors perform a real-data analysis in Section 5, which illustrates how the proposed ITR is different from existing ITRs. However, the authors do not calculate the values of the estimated ITR, so readers cannot compare the performance of the proposed ITR against existing ITRs. Do the authors have estimates of the values of the estimated ITRs?
>
> R: Thank you. We have added a table in the global rebuttal which describes the estimated value function of our proposed ITR and the existing ones. This table further demonstrates the robustness and validity of our method.
>
> W4: The number of treatment-inducing confounding proxies and outcome-inducing confounding proxies were small in both the simulation studies and real-world data analysis. However, the practical appeal of the proximal causal inference framework is its use of proxies for unmeasured confounders, which would suggest the use of many variables as potential proxies. Can the authors include simulations that reflect more realistic settings where more proxy variables are used? How does the proposed method perform as the number of these proxies increases?
>
> R: Following your suggestion, we have revised the real data experiment to include settings with more proxies in the global rebuttal. As we can see, our estimated regime has the largest values throughout and significantly improves upon other regimes including $d_z$ and $d_w$, which is consistent with what we have observed in simulation studies. For the pattern across rows, we note that the optimal values are not expected to be increasing as the number of proxies increases even if they are all compatible with Assumptions 1-5.
>
> Q: How does one do statistical inference for the value of the estimated ITR?
>
> R: Thank you for bringing up this question. We will add the following discussion.
>
> “It may be overly challenging to develop inference results for the value function of the estimated optimal treatment regimes and further studies are warranted. To the best of our knowledge, some recent papers look at the jackknife in applied settings under unconfoundedness (there is some theory in these papers and supplements, though the theory is incomplete):
>
> - X. Jiang, A.E. Nelson, R.J. Cleveland, D.P. Beavers, T.A. Schwartz, L. Arbeeva, C. Alvarez. L.F. Callahan, S. Messier, R. Loeser, M.R. Kosorok (2021). A precision medicine approach to develop optimal exercise and weight loss treatment for overweight and obese adults with knee osteoarthritis. Arthritis Care & Research 73:693-701.
>
> - G. Honvah, H. Cho, M.R. Kosorok (2022). Model selection for survival individualized treatment rules using the jackknife estimator. BMC Medical Research Methodology 22:328.”
>
> L: The authors have not discussed limitations of the work.
>
> R: We will discuss the limitations explicitly in the discussion section. The limitations include the following three aspects.
>
> “We acknowledge several limitations of our work. Firstly, the proximal causal inference framework relies on the validity of treatment- and outcome-inducing confounding proxies. When the assumptions are violated, the proximal causal inference estimators can be biased even if unconfoundedness on the basis of measured covariates in fact holds. Therefore, one needs to carefully sort out proxies especially when domain knowledge is lacking. Secondly, while the proposed regime significantly improves upon existing methods both theoretically and numerically, it is not yet shown to be the sharpest under our considered model. It is still an open question to figure out if a more general policy class could be considered. Thirdly, our established theory provides consistency and superiority of our estimated regime. It is of great interest to derive convergence rates for Proposition 1 and Proposition 2 following Jiang (2017, ICML).”

---

> > ### Comment · Reviewer_of5M · 2023-08-18
> > **Thank you for the response**
> >
> > Thank you for the response. I think the work is now stronger with these additional derivations and experiments. My main concern is that the model class is still somewhat limited, and I will keep my score as is.

---

> > > ### Author Response · Authors · 2023-08-18
> > >
> > > We thank the reviewer’s thorough evaluation and recognition of our response, as well as for sharing insightful suggestions.

---

### Official Review · Reviewer_193J · 2023-07-05

**Soundness:** 4 excellent
**Presentation:** 4 excellent
**Contribution:** 3 good
**Rating:** 7
**Confidence:** 4

**Summary:**

The goal is to learn an optimal individual treatment rule (ITR) where the data suffer from unobserved confounding but where the researcher has a treatment proxy and an outcome proxy.

While the general problem has been studied before by Qi et al (JASA 2023), this paper’s contribution is to broaden the class of ITRs. For a broader class of ITRs, the authors identify the value function and show that it exceeds the value function of the narrower class.


**Strengths:**

My comments are brief because this is a strong paper.

Originality: The essence of the improvement is that, for different covariate values x, one may either use the “outcome” ITR or the “treatment” ITR. This departs from previous work, where either the “outcome” ITR or the “treatment” ITR is used across covariate values.

Quality: The proofs look correct, and the results are easy to interpret. Rates for the objects in Proposition 1 would be an improvement; see the question below.

Clarity: The paper is well written and well referenced.

Significance: The paper contributes to two popular literatures: proxies and ITRs. While its theoretical contribution is modest, it does appear to have practical relevance.

**Weaknesses:**

The theoretical contribution is somewhat incremental.

Take a pass to fix typos, e.g. “netwrok” on line 73.

An extra sentence in Remark 1 would be welcome, that explains the point summarized as “originality” above.


**Questions:**

Is it possible to derive rates for K(pi_hat) and G(pi_bar) in Proposition 1? Or can the authors at least pose this question for future work and give citations of where similar results are derived?

**Limitations:**

Yes

---

> ### Author Rebuttal · Authors · 2023-08-09
>
> We sincerely thank the constructive suggestions from the reviewer.
>
> W1: The theoretical contribution is somewhat incremental.
>
> R: We appreciate the reviewer's feedback and their assessment of our theoretical contribution. We acknowledge the importance of continually advancing the field, and we are trying our best to address the concern of incrementality.
>
> Theorem 2 and Corollary 1 stand as significant achievements in our work, introducing a policy that yields a notably sharper value function within the proximal causal inference framework. The implications of Proposition 1 are also noteworthy, highlighting an excess value bound that signifies the superiority of our optimal ITR when compared to other existing ITRs. To further strengthen our theoretical contributions, we will add an excess value bound (consistency) for the estimated regime in a revised version (this is already presented in the global rebuttal). This supplement will reinforce our theoretical framework.
>
> W2: Take a pass to fix typos, e.g. “netwrok” on line 73.
>
> R: Thank you for pointing out the typo. We will thoroughly review our writing to ensure that typos are minimized.
>
> W3: An extra sentence in Remark 1 would be welcome, that explains the point summarized as “originality” above.
>
> R: Thank you. We will revise the remark following your suggestion.
>
> Q: Is it possible to derive rates for $\mathbb{K}(\hat{\pi})$ and $\mathbb{G}(\bar{\pi})$ in Proposition 1? Or can the authors at least pose this question for future work and give citations of where similar results are derived?
>
> R: Your question is appreciated.  It is possible to derive convergence rates for these two components. For $\mathbb{K}(\hat{\pi})$, we have discussed its asymptotic properties in Appendix G, and to the best of our knowledge, there is some theory in Jiang (2017, ICML) that might be applied when one tries to establish the convergence rate. For $\mathbb{G}(\bar{\pi})$, the convergence rate can be inferred according to our newly proposed Proposition 2 in the global rebuttal and standard treatment regime literature. We will include these topics as an extension of our method in the discussion section.

---

> > ### Comment · Reviewer_193J · 2023-08-21
> >
> > I appreciate the response. I continue to recommend the high rating.

---

> > > ### Author Response · Authors · 2023-08-21
> > >
> > > We thank the reviewer’s acknowledgement of our work and the response.

---

### Official Review · Reviewer_NTDa · 2023-07-06

**Soundness:** 3 good
**Presentation:** 2 fair
**Contribution:** 3 good
**Rating:** 6
**Confidence:** 2

**Summary:**

The paper discusses the optimization of treatment rules in the context of observational data and under assumptions of proximal inference. Various theorems are introduced, and a real data analysis performed using a healthcare example.

**Strengths:**

Below is a list of perceived weaknesses.

The paper is overall sound and the topic of importance. I appreciate the presence of the real data application. Assumptions and results clearly stated.

**Weaknesses:**

Below is a list of perceived weaknesses.

It was not clear to me how the empirical results compare to competing methodological baselines from other approaches (I don't believe the different values presented in the figure represent different algorithmic approaches).

The paper is quite heavy on notation and, at least to me, light on intuitive explanation for findings as they are discussed, limiting insight into the inner workings of why the method works.

I don't know the proximal causal inference literature well so am not well-positioned to discuss the contribution in that subfield of causal inference.

I don't see a discussion of uncertainty estimation in the theoretical or empirical results. Uncertainty estimation in optimized treatment effect regimes can be difficult (e.g., the bootstrap may not be appropriate or may have poor coverage) but may be important to usefulness in practice.

**Questions:**

Is there quantitative evidence that "proximal causes abound in real life scenarios"? The matter would seem dependent on substance area, or knowing this would seem to require access to nature's true DAG.

**Limitations:**

I see no ethnical limitations here.

---

> ### Author Rebuttal · Authors · 2023-08-09
>
> We first thank the reviewer for a careful reading of our work.
>
> W1: It was not clear to me how the empirical results compare to competing methodological baselines from other approaches (I don't believe the different values presented in the figure represent different algorithmic approaches).
>
> R: In Figure 2, we have presented the empirical values regarding different ITRs, including our proposed ITR which results in sharper values, and three ITRs proposed by Qi et al. (2023, JASA) that are baseline methods relying on proximal causal inference framework. For other methods which do not rely on the proximal causal inference framework, we have included a supplementary figure in the global rebuttal, showcasing results derived from causal forest (Wager and Athey, 2018 JASA) and outcome weighted learning (Zhao et al., 2012 JASA).  And we will add these results in a revised version. We also refer to Qi et al. (2023, JASA) for more baseline methods that do not rely on proximal causal inference.
>
> W2: The paper is quite heavy on notation and, at least to me, light on intuitive explanation for findings as they are discussed, limiting insight into the inner workings of why the method works.
>
> R: Sorry for the difficulty in reading caused by the notation. To facilitate your understanding, we summarize the main idea as follows:  we aim at providing the best decision-making strategy for the proximal causal inference framework (Tchetgen Tchetgen et al. 2020; Miao et al. 2018 Biometrika; Cui et al. 2023 JASA), and we focus on comparison with the recently developed methods proposed by Qi et al. (2023, JASA). Technically, on a high level, for each covariate strata $X$, we propose to choose the best decision between the one induced from the outcome confounding bridge $h$ and the one induced from the treatment confounding bridge $q$. We theoretically prove the consistency and superiority of our estimated regime.
>
> W3: I don't see a discussion of uncertainty estimation in the theoretical or empirical results. Uncertainty estimation in optimized treatment effect regimes can be difficult (e.g., the bootstrap may not be appropriate or may have poor coverage) but may be important to usefulness in practice.
>
> R: We will add the following discussion.
>
> “It may be overly challenging to develop inference results for the value function of the estimated optimal treatment regimes and further studies are warranted. To the best of our knowledge, some recent papers look at the jackknife in applied settings under unconfoundedness (there is some theory in these papers and supplements, though the theory is incomplete):
>
> - X. Jiang, A.E. Nelson, R.J. Cleveland, D.P. Beavers, T.A. Schwartz, L. Arbeeva, C. Alvarez. L.F. Callahan, S. Messier, R. Loeser, M.R. Kosorok (2021). A precision medicine approach to develop optimal exercise and weight loss treatment for overweight and obese adults with knee osteoarthritis. -Arthritis Care & Research 73:693-701.
>
> - G. Honvah, H. Cho, M.R. Kosorok (2022). Model selection for survival individualized treatment rules using the jackknife estimator. BMC Medical Research Methodology 22:328.”
>
> Q: Is there quantitative evidence that "proximal causes abound in real life scenarios"? The matter would seem dependent on substance area, or knowing this would seem to require access to nature's true DAG.
>
> R: We agree with the reviewer’s point. The choice of proxies relies on domain knowledge. For example, proxies abound in longitudinal studies as the future can not cause the past, and see Tchetgen Tchetgen et al. (2020) for many other examples. We acknowledge that it is important to correctly identify proxies and the framework when applying our method. Therefore, we add the following limitations of our method in the discussion section.
>
> “We acknowledge several limitations of our work. Firstly, the proximal causal inference framework relies on the validity of treatment- and outcome-inducing confounding proxies. When the assumptions are violated, the proximal causal inference estimators can be biased even if unconfoundedness on the basis of measured covariates in fact holds. Therefore, one needs to carefully sort out proxies especially when domain knowledge is lacking. Secondly, while the proposed regime significantly improves upon existing methods both theoretically and numerically, it is not yet shown to be the sharpest under our considered model. It is still an open question to figure out if a more general policy class could be considered. Thirdly, our established theory provides consistency and superiority of our estimated regime. It is of great interest to derive convergence rates for Proposition 1 and Proposition 2 following Jiang (2017, ICML).”

---

> > ### Comment · Reviewer_NTDa · 2023-08-21
> > **Response to Author Comment**
> >
> > Thank you to the authors for their comments. The various modifications seem to have improved the piece, so I raise my score by one. To bolster the impact of the work, I would emphasize again the importance of comparative baselines against existing methods (to address "even if method X, Y, or Z use different assumptions, perhaps they do better" type questions), uncertainty estimation, and providing clear guidance for applied researchers about how to "know" if they're in the proximal regime.

---

> > > ### Author Response · Authors · 2023-08-21
> > >
> > > We appreciate the reviewer's recognition of our response and valuable suggestions. We will enhance baseline method analysis and discuss uncertainty estimation as well as application guidance in the revision.

---

### Official Review · Reviewer_VJQR · 2023-07-07

**Soundness:** 2 fair
**Presentation:** 2 fair
**Contribution:** 2 fair
**Rating:** 6
**Confidence:** 4

**Summary:**

Most estimation methods for individualized treatment rules (ITRs) assume no unmeasured confounders for valid causal inference. However, such an assumption can be unreasonable, such as when estimating ITRs from observational data. Previous work has applied proximal causal inference to estimate ITRs when this assumption is violated, but is restricted to policy classes that either exclude treatment-inducing confounding proxies or exclude outcome-inducing confounding proxies [1]. To this end, the authors propose estimating a stochastic mixture of both policy classes from [1] to yield a more flexible ITR. Theoretical and simulation results demonstrate the superiority of the proposed method.

**Strengths:**

The assumption of no unmeasured confounders is nearly ubiquitous across ITR estimation methods, despite frequent violations when dealing with observational data. This makes the problem the authors are trying to solve - estimating ITRs when this assumption is violated - very significant. Moreover, the theoretical and simulation results are of sufficient quality to convince me that the method outperforms [1], the existing state-of-the-art in proximal learning.


**Weaknesses:**

While I believe the merits and potential contribution of the paper outweigh its limitations, the theoretical and empirical results of the paper are weaker than that of previous work, and the clarity of the paper can be improved. I go into more detail below:

1. **Theoretical guarantees are much weaker than those of previous methods.** Convergence rates, finite-sample error bounds or asymptotic normality is often derived for ITR methods [2-5], including the method this work seeks to improve over [1]. However, while the authors prove that the proposed method will converge to a policy with better value than that of [1] asymptotically, they do not derive a rate of convergence or establish any finite-sample error bounds. Moreover, the asymptotic analysis from Appendix G assumes convergence of several estimators in $L\_\infty$ space. This is a much stronger assumption than the assumptions made in previous work, which only assumes convergence of estimators in $L\_2$ space (e.g. see Assumption 12 from [1] or Condition B5 from [3]).

2. **The real data analysis does not strengthen the validity of the proposed method.** When applying ITR estimators to real datasets, it is common practice to assess performance by using either (1) an estimator of the value [2-6] or (2) arguments based on domain knowledge that support the validity of the proposed method [1,6]. For example, in [1], the authors argue that the estimated policy is accurate by demonstrating that the estimated coefficients and interpretation of the policy is consistent with findings from previous literature. In contrast, the only conclusions that these authors draw from their real data analysis is that the proposed estimator gives different results than previously proposed methods. Such a conclusion says little about the validity or superiority of the proposed method. One way this analysis could be greatly improved is to look at the patients from Figure 3 where the recommended treatment differs between the proposed method and that of [1], and use domain knowledge or previous literature to argue that the recommendations given by the proposed method is more sound. Alternatively, the authors could make this conclusion by comparing the coefficients of the proposed policy with that of [1].

3. **Empirical comparisons were relatively limited.** The authors only benchmark the proposed method against those of a single previous work, [1]. To conclude that the proposed method achieves state-of-the-art performance, the authors should benchmark the proposed method against additional baselines as well. For example, there are many methods that assume no unmeasured confounders which the authors could evaluate to demonstrate the utility of using a proximal causal inference framework (e.g. [7,8]). There are also other methods that relax the no unmeasured confounders assumption or have shown robustness when such assumptions are violated, such as instrument variable (IV) methods [3,9] and M-learning [4]. How are the assumptions made by proximal learning less restrictive than those made by IV approaches, and can such methods be applied to the simulated datasets? If so, the authors should benchmark the proposed method against these method. And if these methods are not applicable, the authors should explain why in the paper. In addition to the number of competing baselines being limited, the simulated datasets from this work all have the same sample size and behavior policy. When deriving new ITR methods, it is common practice to evaluate the method on datasets of different sample sizes (and if observational data is of interest, varying behavior policies) so as to demonstrate robust performance [2-5,7-11].

4. **The implementation uses very simple estimators.** In theory, $d_z,d_w$ and $\delta$ could be estimated by any weighted classification and regression algorithm. However, in their empirical experiments, the authors only explore estimating $d_z$ and $d_w$ with linear policies and estimating $\delta$ with a Nadaraya-Watson estimator where the bandwidth is chosen based on a heuristic. Moreover, while $h$ and $g$ were estimated using neural networks, the architecture and number of training iterations was fixed a priori. This contrasts to previous works which use more cutting-edge machine learning approaches to estimate the ITR, such as kernel machines, random forests or neural networks, and adopts hyperparameters to the data at-hand using cross-validation [8]. Such works are especially prevalent in top-tier ML conferences [10,11], and better demonstrate broad applicability and flexibility of the proposed method.

5. **Many parts of the paper need to be better written to avoid confusion and address some open questions.** For example, for the real data analysis, it is not clear what assumptions proximal learning is making and how it is useful for the analyzed dataset. It is mentioned that patients were arranged in a "control group" on line 289. Were patients randomized to receive a treatment? If so, wouldn't the no unmeasured confounders assumption hold, as treatment assignment is not being affected by any unmeasured covariates? Also, what is the logic behind the choice of $Z$ and $W$ on line 296 (e.g. why are the variables in $Z$ expected to affect treatment but not outcome), and what specifically are the unmeasured confounders $U$ that we are trying to adjust for? Finally, it is stated that the outcome is censored. Does this mean that 30-day survival rate is censored for some of the patients? It is well-known that optimizing censored outcomes without adjusting for the censoring mechanism can yield bias [5].

>> Here are some other suggestions to reduce points of confusion and improve readability:
a. The explanation of how the proposed method improves upon [1] in the Introduction section (lines 56-65) is confusing. For example, it is stated on line 69-61 that [7] maps an ITR with domain $\mathcal X\times\mathcal W\times\mathcal Z$ with the domain being restricted to $\mathcal X\times\mathcal W$ to $\mathcal X\times\mathcal Z$. While this makes more sense after reading section 2.2, these statements initially appear contradictory. Also on lines 59 and 63 "two optimal in-class ITRs" should be changed to "these two optimal in-class ITRs" to clarify that the authors are referring to the classes mentioned on line 57.
b. The paper has many typos. For example, "Tchetgen Tchetgen et al" in line 34 should include the year and a link to the reference, "netwrok" on line 73 should read "network", and on line 169 the authors should add $V(d_{z\cup w})$ to the argmax.
c. On line 142-143 the authors state that Assumption 3 assumes "Z has sufficient variability with respect to the variability of U". But isn't assumption 3 actually assuming that U has sufficient variability with respect to the variability of Z?
d. On line 173 it is stated that $\mathbb E[Y(a)|X,U]$ may not be identifiable under proximal causal inference. But doesn't assumptions 1-5 allow for such identifiability?
e. Remark 1 is not true. For example if $\pi(X)=0.5$ then $\pi$ is constant but the policy class will not be in $\mathcal D_{\mathcal Z}\cup \mathcal D_{\mathcal W}$.   We actually need the restriction that $\pi$ is both constant and in the set $\\{0,1\\}$.
f. The results of Appendices E and G should appear in the main paper as propositions, theorems or corollaries.
g. For sections where over 10 references are cited back-to-back, I think readability would be improved if these citations appeared in chronological order.
h. The authors should add a Discussion section that summarizes the results of the paper and proposes important avenues for future work (also see my comments on the Limitations section).




References:

1. Qi Z, Miao R and Zhang X. Proximal Learning for Individualized Treatment Regimes Under Unmeasured Confounding. JASA. 2023.

2. Zhao Y, Zeng D, Rush AJ and Kosorok MR. Estimating Individualized Treatment Rules Using Outcome Weighted Learning. JASA 107 (499): 1106-1118. 2012.

3. Qiu H et al. Optimal Individualized Decision Rules Using Instrumental Variable Methods. JASA 116 (533): 174-191. 2021.

4. Wu P, Zeng D and Wang Y. Matched Learning for Optimizing Individualized Treatment Strategies Using Electronic Health Records. JASA 115 (529): 380-392. 2020.

5. Zhao YQ, Zeng D, Laber EB, Song R, Yuan M and Kosorok MR. Doubly robust learning for estimating individualized treatment with censored data. Biometrika 102 (1): 151-168. 2015.

6. Raghu et. al. Continuous State-Space Models for Optimal Sepsis Treatment: a Deep Reinforcement Learning Approach. MLHC 2017.

7. Zhao YQ, Laber EB, Ning Y, Saha S and Sands BE. Efficient Augmentation and Relaxation Learning for Individualized Treatment Rules using Observational Data. JMLR 20: 1-23. 2019.

8. Zhou X, Mayer-Hamblett N, Khan U and Kosorok MR. Residual Weighted Learning for Estimating Individualized Treatment Rules. JASA 112 (517): 169-187. 2017.

9. Pu H and Zhang B. Estimating optimal treatment rules with an instrumental variable: A partial identification learning approach. JRSS-B 83 (2): 318-345. 2021.

10. Yoon J, Jordon J and van der Shaar M. GANITE: Estimation of Individualized Treatment Effects using Generative Adversarial Nets. ICLR 2018.

11. Chen Y, Zeng D, Xu T and Wang Y. Representation Learning for Integrating Multi-domain Outcomes to Optimize Individualized Treatment. NeurIPS 2020.

**Questions:**

Questions that I feel the paper should address and general weaknesses of the paper can be found in the Weaknesses section. As to my suggestions, it may not be feasible to address weaknesses (1) and (4) prior to the camera-ready. However, I would suggest the authors at least acknowledge weakness (1) as a limitation, and discuss how to implement their method in a more data-adaptive manner, even if they don't change the implementation for the empirical experiments. Such a discussion should include how to tune hyperparameters for all functional classes, including estimators of $d_z$, $d_w$, $\delta$, $h$ and $g$.  As to weaknesses (2), (3) and (5), I think it would be feasible to mostly address these weaknesses prior to the camera-ready, and my suggestions on how to address them can be found in the Weaknesses section.

**Limitations:**

I did not notice the authors acknowledge any limitations of the proposed work. I would recommend adding a "Discussion" section that summarizes the results of the work, and addresses limitations by discussing important avenues for future work. In addition to what was discussed in the Weaknesses and Questions sections, another important limitation is that the proposed method assumes that assumptions (1)-(5) hold and that both $h$ and $g$ are correctly specified, while $d_z$ and $d_w$ appear to only require some of these assumptions to hold.

---

> ### Author Rebuttal · Authors · 2023-08-09
>
> We sincerely appreciate the constructive suggestions from the reviewer.
>
> W1: Theoretical guarantees are much weaker than those of previous methods.
>
> R: We express our gratitude for your invaluable suggestions regarding the exploration of convergence rates and finite sample error bounds associated with our proposed estimated ITR. In a revised version, we will enhance the theoretical foundation of our work by including an additional excess value bound (consistency) for the estimated regime (this is already presented in the global rebuttal). And we acknowledge that it is possible to derive convergence rates for the excess value bounds following Jiang (2017, ICML) and our newly proposed Proposition 2 in the global rebuttal. We will include this topic in the discussion section.
>
> Moreover, regarding the assumption we propose in Appendix G, the convergence of estimators in $L_{\infty}$  space is applied as our intention is to guarantee that for each $X$, the conditional expected values based on $\hat{h}$ and $\hat{q}$ are consistent, which is not required by previous literature. Therefore, we employ this assumption to support our subsequent analysis.
>
> W2:  The real data analysis does not strengthen the validity of the proposed method.
>
> R: We sincerely value the constructive feedback provided by the reviewer. As a response, we have incorporated a table in the global rebuttal, detailing the estimated value functions of our proposed ITR, alongside existing approaches. This table further demonstrates the robustness and validity of our method.
>
> W3: Empirical comparisons were relatively limited.
>
> R: We will revise the numerical experiments following your suggestions by including additional baseline methods, varied sample sizes, and altered behavior policies. Due to the space/time limit, we have not presented all these results in the global rebuttal and we will include them in a revised version. As a preview of our progress, we have included a figure in the global rebuttal, showcasing empirical values of different ITRs with a broader spectrum of baseline methods, including ITRs obtained from causal forest (Wager and Athey, 2018 JASA) and outcome weighted learning (Zhao et al., 2012 JASA), as well as existing ITRs mentioned under proximal causal inference framework.
>
> Furthermore, with regard to the IV methods such as those proposed by Cui and Tchetgen Tchetgen (2021, JASA), Pu and Zhang (2021, JRSSB), and Qiu (2021, JASA), it's important to note that these methods operate under the assumption of binary instrumental variables (IVs), while, in our simulation and real data studies, the treatment-inducing confounding proxies $Z$ are not confined to binary values.
>
> W4: The implementation uses very simple estimators.
>
> R: We will include a discussion regarding more machine learning methods for estimating $d_z^*, d_w^*$, and $\delta$ in the supplemental material, as well as the corresponding hyperparameter tuning and cross-validation procedure.
>
> W5: Many parts of the paper need to be better written to avoid confusion and address some open questions.
>
> R: Thank you. We will revise the text carefully to further improve the readability.
>
> Firstly, sorry for the confusion and “control” is a typo. The study is an observational study, and we meant by untreated group.
>
> Secondly, the source of confounding in this study arises from the fact that variables obtained from blood tests may be susceptible to significant measurement errors. Furthermore, besides the lab measurement errors, whether there exists other unmeasured confounding is unbeknownst to the data analyst. Since variables measured from the test offer a singular snapshot of the underlying physiological condition, they hold the potential to act as confounding proxies. Among the ten measures capturing physiological status, a subset of four (pafi1, paco21, ph1, hema1) is selected as they demonstrate a strong connection with both the treatment and the outcome. Then an approach proposed by Tchetgen Tchetgen et al. (2020) ranking proxies according to their strength of association in treatment and outcome models is leveraged to choose the sets of proxies $Z$ and $W$ respectively.
>
> Lastly, as for the censoring problem you mentioned, the censoring in this data is administrative censoring, i.e., everyone is censored at 30 and no one is censored before 30. The outcome of our interest is the survival time up to 30 days, i.e., restricted mean survival time. No censoring bias occurred in this data analysis.
>
> More discussion about the real data application, including assumption verification and sensitivity analysis can be found in Tchetgen Tchetgen et al. (2020), Cui et al. (2023, JASA), Sverdrup and Cui (2023, ICML) and Qi et al. (2023, JASA).
>
> For the additional suggestions put forward in the gray box, we give our responses one by one below.
>
> a. We will revise the text following your suggestion to make it clearer.
>
> b. Thank you. We will check the writing carefully to avoid typos.
>
> c. No. The completeness condition relating the range of $U$ to that of $Z$ states that the set of proxies $Z$ must have sufficient variability relative to the variability of $U$. For more discussion, one can refer to Canay et al. (2013, Econometrika).
>
> d. No. $E[Y(a)|X,U]$ is unidentifiable as we cannot access $U$.
>
> e. Remark 1 is correct as the range of the function $\pi(X)$ is $\{0,1\}$ (defined on line 183). We will revise the text to highlight it.
>
> f. Thank you. We will convert Appendices E and G to formal propositions.
>
> g. We will follow your suggestion to revise the ordering of the citations.
>
> h. We will add a section to discuss limitations and future work (Please find the paragraph in the global rebuttal).
>
> We extend our sincere gratitude for furnishing the reference list, and we will expand our literature review by including the sources you have kindly provided.

---

> > ### Comment · Reviewer_VJQR · 2023-08-15
> > **Response to "Rebuttal by Authors"**
> >
> > I appreciate the authors' thorough rebuttal. Some of my concerns about the paper have been addressed, though other concerns still remain. I go into more detail below:
> >
> > 1. I still feel that the lack of convergence rates and finite-sample error bounds, and the requirement of $L_\infty$ convergence, make the theoretical results  of this work weaker than that of previous work. While the authors proved that their algorithm is consistent in their rebuttal, it seems that their proof also assumes $L_\infty$ convergence via reliance on assumption 6 and Appendix G, and I do not see how this directly leads to a finite-sample error bound for the estimated regime. Moroever, they state that the $L_\infty$ convergence assumption is required to "guarantee that for each $X$, the conditional expected values based on $\hat h$ and $\hat g$ are consistent, which is not required by previous literature." However, [1] also involved estimating similar quantities and managed to avoid $L_\infty$ convergence assumptions. Therefore, I feel it would be possible to avoid such an assumption here as well.
> >
> > 2. While the table of empirical value estimates provided in Table 1 of the authors' rebuttal partially alleviates my concerns regarding the real data analysis, I am still concerned with the way in which performance is being assessed. Specifically, how is value being estimated in Table 1? If the authors are using importance-sampling estimators, these estimators are biased in the presense of unmeasured confounders, which is exactly what this paper is trying to address. Morevoer, if they are using different estimators for different methods, than this makes it difficult to compare value estimates between methods. Therefore, I feel the real data results would be further strengthened by performing a qualitative analysis of the regime estimated by the authors' proposed method and using domain knowledge to validate its performance, similar to what has been done in previous work [1,2,3]. For example, one way to do this is to compare estimated coefficients or treatment recommendations  between the proposed method and that of [1], and argue that the proposed method's estimates are more consistent with previous literature.
> >
> > 3. Between the added comparisons involving additional baselines in Figure 1 of their rebuttal, their clarification on why other baselines such as those based on instrument variables would not be applicable, and their promise to add additional experiments with varying sample sizes and behavioral policies in the revision, all of my concerns related to the weaknesses of the simulated experiments have been satisfied. One remaining suggestion is to clarify in the revision why methods based on IV variables are not applicable to the simulated datasets.
> >
> > 4. I appreciate the authors promising to add discussion on how to combine their methodology with ML algorithms, though I would have liked to see the authors provide such a discussion in their rebuttal. For example, while it is straightforward to apply ML to estimation of $d_z,d_w$ and $\delta$, including tuning hyperparameters, I am curious how the authors would propose tuning hyperparameters to the data at hand when estimating $h$ and $g$, including selecting the network architecture.
> >
> > 5. I appreciate the authors' clarifications regarding the real data set and provided references. I should note that reading [1] did not alleviate my confusion regarding the real dataset. Therefore, I recommend that the authors expand upon the provided clarifications in the revision so that future readers do not suffer from the same confusion. I also appreciate the authors clarifying other points of confusion and promising to add discussion of limitations in the revision.
> >
> >
> > References:
> > 1. Qi Z, Miao R and Zhang X. Proximal Learning for Individualized Treatment Regimes Under Unmeasured Confounding. JASA, 2023.
> >
> > 2. Raghu et. al. Continuous State-Space Models for Optimal Sepsis Treatment: a Deep Reinforcement Learning Approach. MLHC 2017.
> >
> > 3. Luckett DJ et. al. Estimating Dynamic Treatment Regimes in Mobile Health Using V-Learning. JASA, 2019.

---

> > > ### Author Response · Authors · 2023-08-16
> > >
> > > We sincerely express our gratitude to the reviewer for their thorough examination of our rebuttal and for generously sharing invaluable insights. We are genuinely appreciative of the reviewer's acknowledgment of our endeavors in addressing the comments in W3 and W5. We are devoted to enhancing our text in the revision as suggested. To address the reviewer's raised concerns regarding W1, W2, and W4, we will now provide responses below.
> > >
> > > Response to 1:
> > > In contrast to the approach taken by Qi et al. (2023, JASA), whose Theorem 5.1 highlights the consistency of $V(\hat{d}z)$ and
> > >  $V(\hat{d}w)$, our focus, as articulated in our rebuttal, is distinct: to make the value of our proposed regime consistent, we strive to ensure the consistency in the expected values for each stratum $X$, i.e., the consistency of a conditional value function, which is not required by Qi et al. (2023, JASA) or other works within this framework. Therefore, we impose the $L_{\infty}$ convergence assumption.
> > >
> > > We emphasize that the consistency result we have developed is quite novel as the distinct phenomenon of our problem is brand new and different from the previous literature. Regarding the derivation of convergence rates and finite sample error bounds, we recognize that achieving such results is a non-trivial endeavor. Bounding the error metric and establishing advanced theory has captured our keen interest. We will include the discussion of limitations and future work in the revision.
> > >
> > > Response to 2:
> > > We thank you for your insightful suggestion. Our chosen performance evaluation metrics take the latter approach, i.e., using different estimators for different methods. Admittedly, employing such criteria may potentially dilute the persuasiveness of comparisons between methodological value estimates. However, this evaluation is fair to each baseline method, and gives an unbiased estimation under each identification strategy.
> > >
> > > Regarding the inclusion of qualitative analysis, we present an illustrative example below, with plans to expand this discussion in the revision. Notably, the coefficient of cat1_lung is negative with a minor magnitude for $\hat{d}_w$, contrasting with a positive and relatively large coefficient observed for $\hat{d}_z$. These outcomes mirror those outlined in Qi et al. (2023, JASA). This finding suggests that, within the primary disease category of patients with lung cancer, $\hat{d}_z$ advocates for undergoing RHC, while $\hat{d}_w$ displays a notably inconclusive trend. As evidenced by $\hat{\pi}$, the prevailing trajectory for patients with cat1_lung = 1 involves a strong inclination toward undergoing RHC, i.e., $\hat{\pi}=1$, aligning with the guidance offered by $\hat{d}_z$. Significantly, the domain knowledge underscores the potential for patients with advanced lung cancer to develop complications like pulmonary hypertension and coma, potentially warranting RHC for assessing pulmonary vascular changes and informing treatment strategies (Galie et al., 2009). This body of domain-specific knowledge lends support to the recommendations offered by our proposed regime.
> > >
> > > Response to 4:
> > >
> > > We appreciate your attention to hyperparameter tuning and architectural selection during the estimation of bridge functions. Within our experiment, we have diligently conducted both architecture search and hyperparameter optimization, exploring a comprehensive range of configurations. Generally, we consider estimating bridge functions employing multilayer perceptrons with 2-8 fully connected layers with a variable number of hidden units. We then perform a grid search over the following parameters: learning rate (3e-3, 3e-4, 3e-5), $L_2$ penalty coefficient (1e-3, 1e-2, 1e-1), number of epochs (100, 150, 200, 250), batch size (100, 250, 500), depth of network (2, 4, 8), width of network (10, 40, 80). For every permutation of these parameters, we train a network based on the determined architecture and parameter values. Subsequently, we compute the empirical risk, as detailed in Appendix J. Our aim is to pinpoint the parameter combination that yields the lowest empirical risk. These identified optimal parameters are then utilized to construct a refined neural network, which, in turn, serves as the foundation for conducting estimations. For a comprehensive understanding of these intricacies, we suggest delving into supplementary Section B in Kompa et al. (2022, NeurIPS). This supplementary resource offers detailed insights into the specific hyperparameter choices and architectural dimensions.
> > >
> > > References:
> > > Galie N, Hoeper M M, Humbert M, et al. Guidelines for the diagnosis and treatment of pulmonary hypertension: the Task Force for the Diagnosis and Treatment of Pulmonary Hypertension of the European Society of Cardiology (ESC) and the European Respiratory Society (ERS), endorsed by the International Society of Heart and Lung Transplantation (ISHLT)[J]. European heart journal, 2009, 30(20): 2493-2537.

---

> > > > ### Comment · Reviewer_VJQR · 2023-08-17
> > > > **Response to "Official Comment by Authors"**
> > > >
> > > > I thank the authors for elaborating on hyperparameter tuning strategies, and more generally on trying so hard in good faith to address these concerns. Between their initial rebuttal and their most recent reply, many (thought not all) of my concerns now been addressed, and thus I will raise my score of the paper from "borderline accept" to "weak accept". I still have a few remaining suggestions to the authors, which I detail below:
> > > >
> > > > Regarding W1: I believe I now better understand why the authors are making an $L_\infty$ assumption: They are using it to prove consistency of the conditional value function, which is stronger than proving marginal value consistency from Qi 2023. Nonetheless, it stands that without an $L_\infty$ assumption, Qi 2023 still at least achieves marginal value consistency, while this work does not. For the revision, I recommend the authors better explain why they are making an $L_\infty$ assumption in contrast to previous work. I also recommend they try to achieve similar consistency results to Qi 2023 with an $L_2$ assumption, provided doing so would not be that difficult.
> > > >
> > > > Regarding W3: The qualitative analysis provided by the authors is more in-line with what I was hoping for. However, rather than compare the proposed method to $\hat d_w$ using domain knowledge, I recommend the authors instead compare the proposed method to $\hat d_{z\cup w}$ for the revision. Also, when discussing the value table for the real data analysis, I recommend the authors explaining that each method is being evaluated with different methods, and acknowledging that this is a limitation of the comparison.

---

> > > > > ### Author Response · Authors · 2023-08-18
> > > > >
> > > > > We express our sincere gratitude to the reviewer for elevating the score of our work and giving invaluable suggestions that have provided us with a clear direction for our revision.
> > > > >
> > > > > For W1, we will add explanations and limitations of utilizing the $L_\infty$ assumption.
> > > > >
> > > > > For W3, we were comparing the proposed method to $\hat d_z$ and $\hat d_w$ using domain knowledge. For this dataset, note that $\hat d_{z \cup w}$ and $\hat{d}_{z}$ lead to the same regime. Hope this clarifies the concern.
> > > > >
> > > > > Besides, we will acknowledge that each method is being evaluated with different methods, and the values might not be fully comparable when some of the assumptions are not satisfied.

---

> > > > > > ### Comment · Reviewer_VJQR · 2023-08-20
> > > > > > **Response to "Official Comment by Authors"**
> > > > > >
> > > > > > Following up on my previous point on W3, I meant it would make the revision stronger to demonstrate that your proposed method is more consistent with domain knowledge and previous literature than $\hat d_{z\cup w}$, instead of only showing it is superior to $\hat d_w$ as is done in your previous comment.

---

> > > > > > > ### Author Response · Authors · 2023-08-21
> > > > > > >
> > > > > > > Thank you. In the previously presented example, we demonstrated the advantage of $\hat d_z$ for a subgroup of patients with cat1_lung = 1. However, it's important to note that the whole group of patients can be regarded as unions of multiple subgroups based on various distinct features, and the superiority of $\hat d_w$ is evident in some subgroups (e.g., amihx). This shows that our proposed approach offers superior efficacy compared to $\hat d_{z \cup w}$ as our methodology incorporates selection through $\pi$. We will include the discussion in the revision and hope this explanation will address your concern.

---

### Author Rebuttal · Authors · 2023-08-09

We extend our heartfelt gratitude for all the comments provided by the reviewers on our work. Within this global rebuttal, we address recurring concerns raised by multiple reviewers. Our response is organized into three key parts: refining the theoretical analysis, presenting updated experimental results, and introducing a discussion section on limitations.

Part I:
To bolster our theoretical contribution, we establish the consistency of the proposed regime based on the following Assumption 6. Assumption 6 holds for example when $\hat d_z$ and $\hat d_w$ are estimated using indirect methods.

Assumption 6. Suppose that $|E[h(W,\hat{d}_z(X,Z),X)|X]-E[h(W,d_z^*(X,Z),X)|X]|=o_p(n^{-\xi})$ almost surely and $ |E[Yq(Z,A,X)\mathbb{I}\\{\hat{d}_w(X,W) = A\\}|X]-E[Yq(Z,A,X)\mathbb{I}\\{d_w^*(X,W) = A\\}|X]|=o_p(n^{-\varphi})$ almost surely.

Proposition 2. Under Assumptions 1-6, we have $V(\hat{d}_{zw}^{\hat{\pi}})$ converges to

$V(d_{zw}^{\bar{\pi}*})$ in probability.

Proof:

We start with defining four subsets of $\mathcal{X}$, i.e.,

$\mathcal{X}_{g1} = \\{x \in \mathcal{X}: \bar{\pi}(x,\hat{d}_z,\hat{d}_w) = 1, \bar{\pi}(x,d_z^*,d_w^*) = 0\\},$

$\mathcal{X}_{g2} = \\{x \in \mathcal{X}: \bar{\pi}(x,\hat{d}_z,\hat{d}_w) = 0, \bar{\pi}(x,d_z^*,d_w^*) = 1\\},$

$\mathcal{X}_{gc1} = \\{x \in \mathcal{X}: \bar{\pi}(x,\hat{d}_z,\hat{d}_w) = \bar{\pi}(x,d_z^*,d_w^*) = 0\\},$

$\mathcal{X}_{gc2} = \\{x \in \mathcal{X}: \bar{\pi}(x,\hat{d}_z,\hat{d}_w) = \bar{\pi}(x,d_z^*,d_w^*) = 1\\}.$

Then we have

$V(d_{zw}^{\bar{\pi}*}) - V(\hat{d}_{zw}^{\bar{\pi}}) = $

$\int_{x \in \mathcal{X}} \mathbb{E}[\bar{\pi}(X;d_z^*,d_w^*) h(W, d_z^*(X,Z),X) + (1- \bar{\pi}(X;d_z^*,d_w^*)) Y q(Z,A,X)\mathbb{I}\\{d_w^*(X,W) = A\\}$

$- \bar{\pi}(X;\hat{d}_z,\hat{d}_w) h(W, \hat{d}_z(X,Z),X) - (1- \bar{\pi}(X;\hat{d}_z,\hat{d}_w)) Y q(Z,A,X)\mathbb{I}\\{\hat{d}_w(X,W) = A\\}|X = x] f(x)dx$

$=\int_{x \in \mathcal{X}_{g1}} E[Y q(Z,A,X)\mathbb{I}\\{d_w^*(X,W) = A\\} - h(W, \hat{d}_z(X,Z),X)|X=x] f(x)dx $

$+ \int_{x \in \mathcal{X}_{g2}} E[h(W, d_z^*(X,Z),X) - Y q(Z,A,X)\mathbb{I}\\{\hat{d}_w(X,W) = A\\}|X = x] f(x)dx $

 $+ \int_{x \in \mathcal{X}_{gc1}}  E[Y q(Z,A,X)\mathbb{I}\{d_w^*(X,W) = A\} - Y q(Z,A,X)\mathbb{I}\\{\hat{d}_w(X,W) = A\\}|X = x]f(x)dx $

$+ \int_{x \in \mathcal{X}_{gc2}}  E[h(W, d_z^*(X,Z),X) - h(W, \hat{d}_z(X,Z),X)|X = x]f(x)dx.$

Then it is easy to see that
$\int_{x \in \mathcal{X}_{gc1}}  E[Y q(Z,A,X)\mathbb{I}\\{d_w^*(X,W) = A\\} - Y q(Z,A,X)\mathbb{I}\\{\hat{d}_w(X,W) = A\\}|X = x]f(x)dx$

and $\int_{x \in \mathcal{X}_{gc2}}  E[h(W, d_z^*(X,Z),X) - h(W, \hat{d}_z(X,Z),X)|X = x]f(x)dx$ converge to 0 in probability according to Assumption 6.

Therefore, we then essentially need to bound
$\int_{x \in \mathcal{X}_{g1}} E[Y q(Z,A,X)\mathbb{I}\\{d_w^*(X,W) = A\\} - h(W, \hat{d}_z(X,Z),X)|X=x] f(x)dx$.

We further split $\mathcal{X}_{g1}$ to

$\mathcal{X}_{g1,1} = $

$\\{x \in \mathcal{X}_{g1}: E[Y q(Z,A,X)\mathbb{I}\\{d_w^*(X,W) = A\\} - h(W, \hat{d}_z(X,Z),X)|X=x] \in (-C n^{-\zeta},C n^{-\zeta})\\}$

and $\mathcal{X}_{g1,2}=$

$\\{x \in \mathcal{X}_{g1}: E[Y q(Z,A,X)\mathbb{I}\\{d_w^*(X,W) = A\\} - h(W, \hat{d}_z(X,Z),X)|X=x] \notin (-C n^{-\zeta},C n^{-\zeta})\\}$

where $\zeta = \min\\{\xi, \varphi\\}$. Then we can find

$\int_{x \in \mathcal{X}_{g1,1}} E[Y q(Z,A,X)\mathbb{I}\\{d_w^*(X,W) = A\\} - h(W, \hat{d}_z(X,Z),X)|X=x] f(x)dx$

is bounded by $O(n^{-\zeta})$ and $\mathbb{P}(\mathcal{X}_{g1,2})$

converges to 0 in probability based on Assumption 6 and the definition of $\mathcal{X}_{g1}$.

A similar proof can also be conducted to obtain

$\int_{x \in \mathcal{X}_{g2}} E[h(W, d_z^*(X,Z),X) - Y q(Z,A,X)\mathbb{I}\\{\hat{d}_w(X,W) = A\\}|X = x] f(x)$ is small enough.

We then have that $V(\hat{d}_{zw}^{\bar{\pi}})$ converges to

$V(d_{zw}^{\bar{\pi}*})$ in probability.

As we have proved that

$\mathbb{K}(\hat{\pi}) = o(1)$ almost surely in Appendix G,

we finally conclude that $V(\hat{d}_{zw}^{\hat{\pi}})$ converges to

$V(d_{zw}^{\bar{\pi}*})$ in probability.

Part II:
In response to the insightful suggestions provided by the reviewers regarding our numerical experiments and data application, we add a figure and a table in the uploaded PDF file to further demonstrate the robustness and validity of our method.

In detail, for the simulation study, Figure 1 presents empirical values of different ITRs with a broader spectrum of baseline methods, adding causal forest and outcome weighted learning. As we can see, the proposed method consistently achieves the largest value.

For the real data analysis, Table 1 describes the estimated value functions of our proposed ITR, alongside existing approaches, and we also include settings with increasing numbers of proxies. As we can see from the table, the proposed regime has the largest value among all settings.

Part III:
We will add a discussion section to address limitations, and discuss potential directions for future work:

“We acknowledge several limitations of our work. Firstly, the proximal causal inference framework relies on the validity of treatment- and outcome-inducing confounding proxies. When the assumptions are violated, the proximal causal inference estimators can be biased even if unconfoundedness on the basis of measured covariates in fact holds. Therefore, one needs to carefully sort out proxies especially when domain knowledge is lacking. Secondly, while the proposed regime significantly improves upon existing methods both theoretically and numerically, it is not yet shown to be the sharpest under our considered model. It is still an open question to figure out if a more general policy class could be considered. Thirdly, our established theory provides consistency and superiority of our estimated regime. It is of great interest to derive convergence rates for Proposition 1 and Proposition 2 following Jiang (2017, ICML).”

Reference:
H. Jiang. Uniform convergence rates for kernel density estimation. ICML, 2017.

---

### Decision · Program_Chairs · 2023-09-21

**Decision:**

Accept (poster)

**Comment:**

This is a technically solid paper with a tangible contribution to the causal literature, tying together in a useful and non-trivial way recent results from proximal causal inference and from the literature learning individual treatment rules. Reviewers found the both the theoretical analysis and experiments to be of overall good quality, with some issues resolved during the discussion period.